# Time Series Representations with Hard-Coded Invariances

**Thibaut Germain** [*1] **Chrysoula Kosma** [*1] **Laurent Oudre** [1]

## Abstract

Automatically extracting robust representations from large and complex time series data is becoming imperative for several real-world applications. Unfortunately, the potential of common neural network architectures in capturing invariant properties of time series remains relatively underexplored. For instance, convolutional layers often fail to capture underlying patterns in time series inputs that encompass strong deformations, such as trends. Indeed, invariances to some deformations may be critical for solving complex time series tasks, such as classification, while guaranteeing good generalization performance. To address these challenges, we mathematically formulate and technically design efficient and hard-coded *invariant convolutions* for specific group actions applicable to the case of time series. We construct these convolutions by considering specific sets of deformations commonly observed in time series, including *scaling*, *offset shift*, and *trend*. We further combine the proposed invariant convolutions with standard convolutions in single embedding layers, and we showcase the layer's capacity to capture complex invariant time series properties in several scenarios.

## 1. Introduction

Recently, there has been a growing interest in applying machine learning to time series data, further necessitated by the challenging properties of time series, including varying modalities, high noise levels, and distribution shifts.

Machine learning for time series has gradually moved from statistical methods, e. g., autoregressive models (Box et al.,

*Equal contribution [1]Université Paris-Saclay, Université Paris Cité, ENS Paris Saclay, CNRS, SSA, INSERM, Centre Borelli, F-91190, Gif-sur-Yvette, France. Correspondence to: Thibaut Germain <thibaut.germain@ens-paris-saclay.fr>, Chrysoula Kosma <chrysoula.kosma@ens-paris-saclay.fr>.

*Proceedings of the 42nd International Conference on Machine Learning*, Vancouver, Canada. PMLR 267, 2025. Copyright 2025 by the author(s).

2015) and dictionary-based classification (Middlehurst et al., 2019), to neural networks. Notable architectures are built upon recurrent layers, convolutional layers, and, more recently, transformers. More specifically, convolutional neural networks (CNNs) consistently lead time series tasks such as classification and clustering (Ismail Fawaz et al., 2019; Tonekaboni et al., 2021). Often integrated with other modules, CNNs excel in feature extraction, while offering interpretability through kernel weight visualization.

Advances in invariance modeling for deep learning have been achieved with proper mathematical formalism of group action on data, notably images and graphs (Kvinge et al., 2022; Bronstein et al., 2021). Two strategies for incorporating invariance into deep networks are learning them through data augmentation, such as contrastive learning (Antoniou, 2017) or hard-coding invariances directly within the network architecture. Translation in CNNs and permutation in graph neural networks (GNNs) are two prominent examples of hard-coded invariances in terms of architectural design, among others (Bietti & Mairal, 2019; Horie et al., 2020). Relevant works range across sets, images, point clouds, and graphs (Zaheer et al., 2017; Keriven & Peyré, 2019).

Incorporating knowledge on time series invariances during training is an emerging field of study. Several works focus on learning invariances, e. g., contrastive learning that leverages augmentations and customizable losses (Franceschi et al., 2019; Eldele et al., 2021). Yet, the selection of views within the time series collection to contrast, as well as the types of transformations to consider (e. g., scaling, shifting), is often arguable within the research community (Yue et al., 2022). While contrastive learning constitutes an implicit way of introducing invariances into learning, very few studies leverage time series hard-coded invariances. In this work, we aim to embed invariances into the network design, similar to approaches that achieve permutation invariance in graphs (Maron et al., 2018) and local translation and rotation invariance in images (Weiler & Cesa, 2019). Such invariant networks offer improved generalization properties compared to their learned counterparts at the expense of additional computational costs (Kvinge et al., 2022).

Enforcing deformation-specific invariances in network design can significantly improve performance for time series tasks. For example, the trend in time series constitutes a

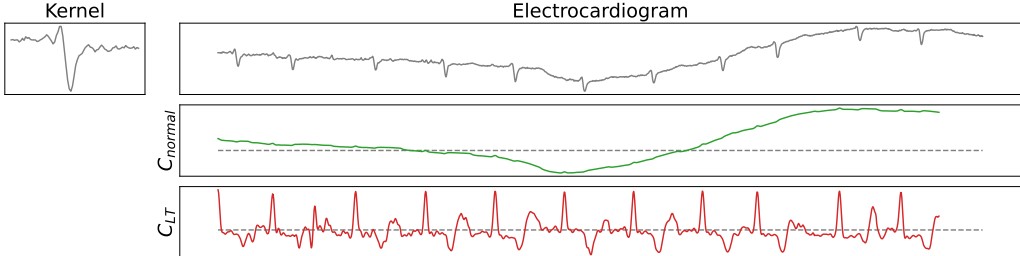

Figure 1: **Top right:** Segment of a electrocardiogram (ECG) from the MIT-BIH dataset (Goldberger et al., 2000; Moody & Mark, 2001). **Top left:** The convolutional kernel is the first heartbeat. **Middle:** With the normal convolution, individual heartbeats are not identifiable as they are blurred by the trend. **Bottom:** With linear trend invariant convolution, all heartbeat occurrences are identifiable as they are positively correlated with the kernel, minimizing deformations induced by the trend.

common deformation, and removing its effects is an active research area known as baseline removal (Hippke et al., 2019; Zhang et al., 2010). Modern time series machine learning pipelines often perform baseline removal as a pre-processing step (Baek et al., 2015; Yan et al., 2019; Zhang et al., 2020b). However, these methods usually require extensive hyper-parameter tuning (Zhang et al., 2020a; Singhal et al., 2020). In deep learning, this challenge has been tackled through trend modeling components (Oreshkin et al., 2019), which has also been combined with contrastive learning (Liu et al., 2024; Woo et al., 2022a). Interestingly, convolution-based networks can leverage hard-coded invariant filters to accurately approximate trend invariance. Indeed, convolutions focus on local information, and similarly to spline functions, the trend can locally be approximated by low-degree polynomial functions, which, in its simplest form, can be seen as linear (1-degree). Figure 1 illustrates the cross-correlation between a single heartbeat and an ECG affected by a trend assuming a standard convolution and a convolution invariant to a linear trend. Surprisingly, the standard convolution fails to detect the correlation between the query heartbeat and the individual ones. On the contrary, the linear trend invariant convolution successfully identifies correlations to all underlying heartbeats and potentially offers more robustness for any ECG diagnosis (Liu et al., 2021b).

Following the previous observations, our work tackles the challenge of hard-coded spatiotemporal invariance within convolutional layers for time series accounting for deformations like offset shift or linear trend and extending beyond mere time invariance. In the literature, very few works deal with hard-coded time invariance, and most are concerned with local time-warping invariance (Shulman, 2019) or time rescaling invariance (Jacques et al., 2022). In addition, our mathematical framework provides an exact formulation for invariant convolutions, as opposed to approximating invariance in previous works. More specifically:

**Section 3**. We formulate time series deformations via group

actions and introduce invariance under these actions. We then design generic and exact hard-coded invariant convolutions capable of handling deformations such as trends.

**Section 4.1**. We highlight the sensitivity of standard convolution to common deformations and demonstrate how deformation-invariant convolutions mitigate this issue, achieving better generalization than learned invariance.

**Section 4.2, 4.3 & Appendix A.5.1**. We showcase the effectiveness of the proposed convolutions against state-of-the-art baselines across multiple tasks. The results demonstrate that our hard-coded invariant convolutions offer a fast, generalizable, and robust approach to time series representation learning.

## 2. Related Work

**Deep Learning for Time Series.** Dominant deep learning frameworks for time series leverage multi-layer perceptrons (MLPs) (Oreshkin et al., 2019), convolutional networks (CNNs) (Bai et al., 2018) and recurrent ones (RNNs) (Salinas et al., 2020), as well as transformer-based networks (i. e., built upon the attention mechanism) (Wen et al., 2022). Convolutional kernels have traditionally dominated feature extraction in time series, from shapelets (Ye & Keogh, 2011) to the recently successful ROCKET (Dempster et al., 2020), that exploits several random kernels. Additionally, convolutional layers of different kernel sizes, often stacked in deep architectures (Ismail Fawaz et al., 2019), with increased receptive fields, such as INCEPTIONTIME (Ismail Fawaz et al., 2020) and RESNET (Wang et al., 2017), are prominent for time series classification. Similarly, TIMESNET (Wu et al., 2022) model capitalizes on convolutional layers to capture variations of multiple periodicities of 2D transformed multivariate time series, to solve multiple time series tasks. Beyond standard CNNs for time series, T-WaveNet (Minhao et al., 2021) is a tree-structured wavelet neural network that decomposes the input signal into various frequency subbands with similar energies based on

the dominant frequency range. Another recent hierarchical CNN-based model for time series forecasting, SCINET (Liu et al., 2022), repeatedly downsamples and convolves the input to enable information sharing at several resolutions. Furthermore, leveraging the success of the attention mechanism in text, transformer-based architectures have lately proven successful in capturing temporal interactions between multivariate time series inputs, mainly in time series forecasting (Zhou et al., 2021; 2022; Woo et al., 2022b; Liu et al., 2021a). For instance, AUTOFORMER (Wu et al., 2021) combines decomposition modules with an auto-correlation in place of self-attention, while CROSS-FORMER (Zhang & Yan, 2022) capitalizes on 2D vector array embeddings that preserve temporal and channel information, followed by temporal and channel-wise cross-attention modules. By merging attention with masked autoencoders, UP2ME (Zhang et al., 2024) applies univariate pre-training followed by multivariate cross-channel fine-tuning to generalize across multiple tasks. In parallel, several recent works, evaluate forecasting architectures also on the anomaly detection (Xu, 2021), by reformulating the task to point-wise reconstruction, with reconstruction error being the anomaly criterion. To overcome the sensitivity of transformers in overfitting, recent MLP-based architectures have showcased superior performance in forecasting, e. g., TSMIXER (Chen et al., 2023), TIMEMIXER (Wang et al., 2024a), FRETS (Yi et al., 2024), while several studies doubt the robustness of the former in time series modeling (Zhang et al., 2022; Zeng et al., 2023). Notably, forecasting methods often perform trend-seasonal decomposition with moving averages, such as PERI-MIDFORMER (Wu et al., 2024), which applies self-attention between multiple periodic features. However, moving average kernels mainly address offset shifts, contrary to our more principled and flexible approach to deformation invariance (e. g., linear or higher-order trends).

**Invariances for Time Series Modeling.** Besides supervised methods, multiple approaches emphasize unsupervised learning for extracting representations from time series data before the downstream task (Nie et al., 2022; Dong et al., 2024). Among prominent approaches, self-supervised contrastive methods enforce invariances between representations by applying transformations (e. g., scaling, shifting, or noise injection) to the input and training the model to map them to the same underlying representations (Chen et al., 2020). Typically, CNNs constitute basic blocks for various time series augmentation-based contrastive frameworks, such as TS-TCC (Eldele et al., 2021) and TIMECLR (Yang et al., 2022). Except for transformation-based augmentations, samples can also be contrasted with sampled subseries (Franceschi et al., 2019), adjacent segments (Tonekaboni et al., 2021), or a combination of all (Yue et al., 2022). Unlike deep learning, time series invariances have long been a central focus in classical time series data mining approaches

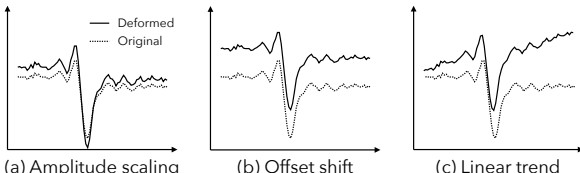

Figure 2: Deformations applied to an example series, including amplitude scaling, offset shift, and linear trend.

(Esling & Agon, 2012). For instance, local time warping invariance (Ding et al., 2008) can be tackled by dynamic time warping (DTW). Traditionally, amplitude and offset invariances are accomplished by Z-normalizing the data (Paparrizos et al., 2020). However, while Z-normalization removes global offsets, it fails to address local distortions. The LT-normalized distance (Germain et al., 2024b) extends Z-normalized distance by modeling invariance to linear trends. Diffeomorphisms have also been leveraged for shift invariance in neural networks through a differentiable bijective function mapping between time series manifolds (Demirel & Holz, 2025). Closely related, authors in (Germain et al., 2024a) propose methods to quantify deformations like time-warping through modeling temporal deformations as diffeomorphisms acting on time series.

## 3. Method

In real-world signals, constant baseline shifts, slow drifts and long-term changes can arise by external factors and distort underlying signal patterns. Therefore, amplitude scaling, offset shift, and linear trends can be identified as common time series deformations, as depicted in Figure 2 on an ECG heartbeat. These deformations can serve as examples for the empirical validation of our broader theoretical framework of time series invariant convolutions. The proposed framework for learning time series representations with hard-coded invariances consists of two main components. The first component is a *group action* that formalizes how certain deformations transform time series (Section 3.1), resulting in their deformed counterparts. Deformed time series are observable in practice, due to noise or trends. To ensure invariance to specific deformations, that can be important for many applications, the second component is a *mapping function* that constructs embeddings of deformed time series while remaining invariant to such deformations (Section 3.1) Finally, these embedding maps can be efficiently integrated into convolutional operations (Section 3.2) to extract robust local features to non-informative deformations.

### 3.1. Invariant embedding for time series

We, next, construct an embedding invariant to a predefined set of deformations. Essentially, the embedding is expected

to map a geometrical object or any of its deformed versions to the same representative. In addition, we present some important properties that should verify the embedding and tailor the proposed framework to the case of time series.

**Deformations and group action.** From a geometrical viewpoint, the notion of invariance depends on the representation of deformations and the definition of the action of a deformation on a geometrical object. A classical approach consists of representing a deformation as an element of a group and its action by a group action:

**Definition 3.1** (Group action). A group $G$ with neutral $e$ acts on the left on a set $M$, if there exists a map $a : G \times M \mapsto M$ that verifies:

1) $a(e, m) = m, \quad \forall m \in M$
2) $a(g, a(h, m)) = a(gh, m), \quad \forall (g, h) \in G^2, \forall m \in M.$

To simplify notation, the left action of $g \in G$ on $m \in M$ is denoted $g \cdot m$. For a group $G$ that acts on the left on a set $M$, the orbit of $m \in M$ is the set of all its deformed versions $[m] = \{g \cdot m \mid g \in G\}$. The set of independent orbits, denoted $M/G$, is called quotient space, and if this set is reduced to a singleton, the action of $G$ on $M$ is said transitive, and it verifies that for any $m \in M$ its orbit is the whole set: $[m] = M$.

**A group action for time series.** Leveraging measure theory, we model the set of time series by the Hilbert space $L^2(I, \mathbb{R}^D, \mu)$ of functions defined on the closed interval $I \subset \mathbb{R}$ taking value in $\mathbb{R}^D$ and square-integrable for the Borel measure $\mu$. The inner product on $L^2(I, \mathbb{R}^D, \mu)$ is defined as:

$$\langle f, g \rangle_L = \int_I \langle f(t), g(t) \rangle d\mu(t) \tag{1}$$

where $\langle ., . \rangle$ is the dot product on $\mathbb{R}^D$. Let $H$ be a finite dimensional vector subspace of $L^2(I, \mathbb{R}^D, \mu)$, we model the group of deformations as the set $\mathbb{R}_+^* \ltimes H$ with the composition rule $(\lambda_2, h_2) \times (\lambda_1, h_1) = (\lambda_2 \lambda_1, h_2 + \lambda_2 h_1)$. Finally, we model the group action by the application:

$$\begin{aligned} (\mathbb{R}_+^* \ltimes H) \times L^2(I, \mathbb{R}^D, \mu) &\rightarrow L^2(I, \mathbb{R}^D, \mu) \\ ((\lambda, h), f) &\mapsto \lambda f + h \end{aligned} \tag{2}$$

This is a general group action that is not transitive as $H$ is a finite-dimensional vector subspace of $L^2(I, \mathbb{R}^D, \mu)$. By convention, we refer to $\mathbb{R}_+^* \ltimes H$ as the set of rigid deformations. The customization of the group action depends on the choice of basis for the subspace $H$. For instance, the Z-normalization (Paparrizos et al., 2020) is an invariant offset shift which corresponds to the subspace of deformations $\{h : I \mapsto c \mid c \in \mathbb{R}^D\}$ with the basis $\{h_i : I \mapsto e_i / \sqrt{length(I)} \mid i \in [1, \dots, D]\}$ where $(e_i)_{i \in [1, \dots, D]}$ is the orthonormal basis of $\mathbb{R}^D$.

**Invariant embedding.** An embedding invariant to a group action is expected to map any element of an orbit to the same representative, and it is defined as follows:

**Definition 3.2** (Invariant & orbit-injective embedding). An embedding map $L : M \mapsto N$ is said to be $G$-invariant, if for any $(g, m) \in G \times M$, $L(g \cdot m) = L(m)$. Additionally, $L$ is said to be orbit-injective if the application $\tilde{L} : [m] \in M/G \mapsto L(m) \in N$ is injective.

Note that an invariant embedding is meaningful in the case of a non-transitive group action. In addition, if the embedding is orbit-injective, each orbit has a distinct representative.

For now, we focus on the action of the finite-dimensional subspace $H$ of a Hilbert space $M$ by the usual vector addition: $(h, m) \in H \times M \mapsto m + h \in M$. The following proposition exhibits a $H$-invariant embedding that is also orbit-injective.

*Proposition* 1. Let $P_H$ be the orthogonal projector on $H$, and $I_d$ be the identity map on $M$, the embedding, $L = I_d - P_H$ (the projector on $H^\perp$) is $H$-invariant and orbit-injective.

*Proof.* See Appendix A.1. □

*Remark* 3.3. If $(h_i)_{i \in [\![1, N]\!]}$ is an orthonormal basis of the finite dimensional vector subspace $H$, then the orthogonal projector on $H$ as an explicit formulation $P_H : m \in M \mapsto \sum_{i=1}^N \langle m, h_i \rangle_L h_i \in H$.

Invariance to amplitude scaling can easily be incorporated in an embedding defined by the previous proposition:

*Proposition* 2. Let $L : M \mapsto M$ be the $H$-invariant and orbit-injective embedding map induced by the orthogonal projector on $H$ as defined in proposition 1. The embedding map:

$$\hat{L} : m \in M \mapsto \begin{cases} L(m)/\|L(m)\|_M & \text{if } m \in M \backslash H \\ 0_M & \text{else} \end{cases} \tag{3}$$

is $(\mathbb{R}_+^* \ltimes H)$-invariant and orbit-injective.

*Proof.* $(\mathbb{R}_+^* \ltimes H)$-invariance is due to the linearity and $H$-invariance of $L$, and the orbit-injectivity in induced by the linearity and orbit-injectivity of $L$. □

**An example: the univariate Z-normalization.** We are looking for an embedding invariant to amplitude scale and offset shift in the case of univariate discrete time series. The set of time series is modeled by $L^2([0, l], \mathbb{R}, \mu)$ where $l \in \mathbb{N}^*$, $\mu = \sum_{i=1}^l \delta_i$ and $\delta_i$ is the dirac measure at $i$. The set offset shifts is the subspace generated by the unit norm function $e : t \in [0, l] \mapsto 1/\sqrt{l} \in \mathbb{R}$. According to Proposition 2 the invariant embedding of a non-constant function $f$ is the function: $(f - \langle f, e \rangle_L e)/\|f - \langle f, e \rangle_L e\|_L$ which

leads to $(f(i) - \mu_f)/(\sqrt{l}\sigma_f)$ where $\mu_f = l^{-1} \sum_{i=1}^{l} f(i)$ and $\sigma_f^2 = l^{-1} \sum_{i=1}^{l} (f(i) - \mu_f)^2$.

## 3.2. Invariant convolution

CNNs have been successful in many applications related to time series, essentially becoming a key building block of the latest deep neural networks. Their success comes from their ability to capture local information in long time series. However, CNNs remain sensitive to some deformations like amplitude scaling or offset shifts (Mallat, 2016). In this section, we propose a novel convolution that is invariant to rigid deformations at a local scale while remaining computationally efficient.

**The formalism.** Let $\mathsf{L}_{loc}^2(\mathbb{R}, \mathbb{R}^D, \mu)$ be the set of signals, we assume that the signal in square integrable on any compact of $\mathbb{R}$. Let $\mathsf{L}^2(\mathsf{I}, \mathbb{R}^D, \mu)$ be the set of kernels where $\mathsf{I} \subset \mathbb{R}$ is a closed interval. The classical convolution layer, named 1D-CNN, between a signal $f$ and a kernel $w$ is the signal:

$$f * w : u \in \mathbb{R} \mapsto \int_\mathsf{I} \langle f(u+t), w(t) \rangle d\mu(t) \in \mathbb{R} \quad (4)$$

Let assume a group of rigid deformations $\mathsf{G}$ acting on $\mathsf{L}^2(\mathsf{I}, \mathbb{R}^D, \mu)$, and $\hat{L}$ the $\mathsf{G}$-invariant embedding map defined by Proposition 2. For any $u \in \mathbb{R}$, we can define the operator $K_u^\mathsf{G}$ that maps the restriction of any signal $f$ on the closed interval $u + \mathsf{I}$ to its $\mathsf{G}$-invariant representative:

$$K_u^\mathsf{G} : \left| \begin{array}{ccc} \mathsf{L}_{loc}^2(\mathbb{R}, \mathbb{R}^D, \mu) & \to & \mathsf{L}^2(\mathsf{I}, \mathbb{R}^D, \mu) \\ f & \mapsto & \hat{L}\left(t \in \mathsf{I} \mapsto f(t+u)\right) \end{array} \right. \quad (5)$$

Leveraging these operators we define the $\mathsf{G}$-invariant convolution between a signal $f$ and a kernel $w$ as the signal:

$$f *^\mathsf{G} w : u \in \mathbb{R} \mapsto \int_\mathsf{I} \langle (K_u^\mathsf{G} f)(t), w(t) \rangle d\mu(t) \in \mathbb{R} \quad (6)$$

**Fast computation.** For a group of rigid deformations $(\mathbb{R}_+^* \ltimes \mathsf{H})$ with $(h_i)_{i \in [\![1,N]\!]}$ a basis of $\mathsf{H}$, thanks to Remark 3.3, the inner product between the invariant representation of $f \in \mathsf{L}^2(\mathsf{I}, \mathbb{R}^D, \mu)$ and $w$ can be decomposed as follows: $\langle \hat{L}(f), w \rangle_\mathsf{L} = (\langle f, w \rangle_\mathsf{L} - \sum_{i=1}^{N} \langle f, h_i \rangle_\mathsf{L} \langle w, h_i \rangle_\mathsf{L})/\|L(f)\|_\mathsf{L}$. Assuming discrete signals, the computation of $\langle \hat{L}(f), w \rangle_\mathsf{L}$ requires the computation of $2N + 2$ dot products. However, convolving a batch of $B$ signals of length $L$ with the kernel, the number of inner products to compute drops from $BL(2N + 2)$ to $BL(N + 2) + N$ as the inner products between the kernel and the basis are shared across signals and subsequences. It leads to the time complexity $\mathcal{O}(BLNCW)$ where $C$ is the number of channels, $W$ is the kernel size and assuming

that $N << L$. Invariant convolutions do not consider small-size kernels (2 or 3 timestamps) but rather large kernels (30 or more). The traditional approach to convolution is not tractable in such a context. Instead, we leverage the Fast Fourier transform (FFT) (Mathieu et al., 2013), which changes the time complexity to $\mathcal{O}(BNCL \log(L))$. The computational time is identical for any window size, as the computation with the FFT does not depend on the kernel size. In the experimental results, we show that our proposed invariant convolutions benefit from fast computation.

## 4. Experimental Evaluation

We present an extended experimental evaluation illustrating the use and performance of hard-coded invariant convolutions. As already presented in Section 3 (Figure 2), we specifically focus on convolutions invariant to constant functions (offset shift) $\mathsf{H}_{off} = \{t \in \mathsf{I} \mapsto b \mid b \in \mathbb{R}^D\}$ or affine functions (linear trend) $\mathsf{H}_{LT} = \{t \in \mathsf{I} \mapsto at + b \mid (a,b) \in \mathbb{R}^D \times \mathbb{R}^D\}$. Based on the discussion in the Introduction, these deformations are the simplest local approximation of trend deformation. In addition, we also include invariance to amplitude scaling, which is a common source of inter-individual variability. The experimental evaluation is organized as follows:

**I. Robustness to Deformations.** We experimentally prove the robustness of the proposed invariant convolutions compared to vanilla ones and contrastive-based methods on a classification task when considering deformations.

**II. Classification Performance.** We show the competitive performance of an example architecture built upon a pool of normal and invariant convolutions on classification for several benchmark datasets. We also perform ablations and computational efficiency studies for the proposed method.

**III. Generalization Performance.** We assess the robustness of the proposed invariant convolutions on a transfer learning classification experiment against relevant contrastive learning methods.

**IV. Anomaly Detection Performance.** We finally capitalize on invariant filters to perform reconstruction-based anomaly detection by introducing an example decoder that effectively combines different types of features to reconstruct the signal. The experimental protocol and the results are presented in Appendix A.5.1, revealing the benefit of invariant convolutions.

**Code and Experimental Details.** The source code for this work is available on GitHub[1]. More information about the datasets and the implementation details for the proposed method and baselines can be found in Appendix A.3 and A.4, respectively.

---

[1] https://github.com/sissykosm/TS-InvConv

Table 1: Robustness study of distinct convolutional filter types with respect to invariance, i. e., standard ones (non-invariant), offset invariant, and trend invariant, on five *UCR* datasets. Models are trained on *normalized* data and tested on four additional synthetic deformation scenarios (random offset (off.), random linear trend (LT), smooth random walk (RW) and their combinations). Higher is better, best methods in **bold**, second best underlined. The self-supervised TS-TCC method is pre-trained with all deformation augmentations, and it is then fine-tuned (FT) on the raw normalized data. For the three convolutional variants, knowledge of the distribution of deformations is not incorporated during training, and thus, the model is evaluated in an out-of-distribution setting.

| | | CONV
- normal - | INV. CONV
- offset - | INV. CONV
- trend - | TS-TCC (FT) (2021)
- offset, trend - |
|---|---|---|---|---|---|
| *HandOutlines* | *Normalized* | 77.84 ± 0.0 | 72.43 ± 0.54 | 70.54 ± 0.71 | **88.96 ± 1.17** |
| | *+ off.* | 41.08 ± 0.47 | **72.43 ± 0.54** | 70.72 ± 1.02 | 47.83 ± 0.54 |
| | *+ LT* | 37.39 ± 0.16 | **71.08 ± 1.08** | 70.90 ± 0.82 | 40.87 ± 1.41 |
| | *+ off., LT* | 35.68 ± 0.47 | 70.45 ± 1.22 | **70.63 ± 0.41** | 37.07 ± 0.87 |
| | *+ off., RW* | 35.95 ± 0.0 | 62.34 ± 1.02 | **64.59 ± 1.24** | 35.82 ± 0.15 |
| *MixedShapesRegularTrain* | *Normalized* | **93.91 ± 0.39** | 88.48 ± 1.79 | 92.26 ± 0.70 | 93.44 ± 1.51 |
| | *+ off.* | 38.96 ± 0.99 | 88.41 ± 1.83 | 92.24 ± 0.73 | **92.56 ± 1.31** |
| | *+ LT* | 29.33 ± 2.16 | 84.63 ± 3.92 | **91.55 ± 0.69** | 82.96 ± 4.72 |
| | *+ off., LT* | 28.22 ± 1.49 | 84.05 ± 4.08 | **91.64 ± 0.80** | 81.35 ± 4.08 |
| | *+ off., RW* | 22.94 ± 0.71 | 76.98 ± 4.62 | **90.59 ± 0.64** | 68.56 ± 4.86 |
| *NonInvasiveFetalECGThorax1* | *Normalized* | **91.35 ± 0.05** | 86.53 ± 0.06 | 85.00 ± 0.33 | 84.19 ± 1.18 |
| | *+ off.* | 15.32 ± 0.44 | **85.11 ± 0.73** | 83.84 ± 0.12 | 71.47 ± 6.51 |
| | *+ LT* | 8.09 ± 0.23 | 42.49 ± 0.33 | **82.56 ± 0.45** | 49.54 ± 5.97 |
| | *+ off., LT* | 6.09 ± 0.17 | 42.36 ± 0.65 | **82.02 ± 0.52** | 47.61 ± 6.70 |
| | *+ off., RW* | 3.48 ± 0.14 | 37.22 ± 0.94 | **73.33 ± 0.05** | 31.55 ± 6.16 |
| *FordB* | *Normalized* | 82.35 ± 0.57 | **83.33 ± 0.5** | 82.84 ± 0.65 | 78.37 ± 0.30 |
| | *+ off.* | 60.91 ± 3.17 | 81.61 ± 1.38 | **83.37 ± 0.31** | 78.29 ± 0.43 |
| | *+ LT* | 53.83 ± 1.10 | 70.50 ± 6.02 | **82.68 ± 0.91** | 76.85 ± 2.23 |
| | *+ off., LT* | 53.04 ± 0.75 | 69.47 ± 7.34 | **82.35 ± 0.66** | 76.49 ± 2.83 |
| | *+ off., RW* | 50.91 ± 0.26 | 68.80 ± 8.21 | **80.37 ± 0.43** | 75.77 ± 1.67 |
| *Yoga* | *Normalized* | 76.12 ± 0.24 | 76.80 ± 1.69 | 76.92 ± 0.14 | **80.64 ± 0.63** |
| | *+ off.* | 53.73 ± 0.26 | 72.07 ± 1.93 | **75.71 ± 0.18** | 74.97 ± 1.59 |
| | *+ LT* | 52.86 ± 0.14 | 67.71 ± 2.99 | **74.17 ± 0.29** | 62.21 ± 0.74 |
| | *+ off., LT* | 51.13 ± 0.07 | 64.85 ± 2.05 | **72.88 ± 0.38** | 61.85 ± 1.26 |
| | *+ off., RW* | 48.82 ± 0.12 | 65.67 ± 0.74 | **72.58 ± 0.19** | 62.69 ± 2.64 |
| **Percentage Drop (%)**
*with respect to Normalized*
*(lower the better in abs. value)* | *+ off.* | -48% ± 20% | -1% ± 2% | **0% ± 0%** | -13% ± 17% |
| | *+ LT* | -55% ± 22% | -16% ± 17% | **-1% ± 1%** | -26% ± 19% |
| | *+ off., LT* | -57% ± 22% | -18% ± 17% | **-1% ± 2%** | -28% ± 20% |
| | *+ off., RW* | -59% ± 23% | -23% ± 16% | **-6% ± 4%** | -34% ± 22% |

## 4.1. Robustness to Deformations

**Protocol.** We aim to evaluate the robustness of trend deformations of hard-coded invariant convolutions (scaling/offset & scaling/linear trend) compared to vanilla convolutions and learned invariant representations with contrastive learning. Robustness is evaluated on a classification task. We consider 5 datasets from the *UCR* archive (Dau et al., 2019), which, by default, are all Z-normalized. Concerning baselines, we include three versions of inception-like (Ismail Fawaz et al., 2020) networks (a single convolution layer with kernels of four different sizes followed by a linear classifier) whose architecture is presented in Appendix A.2. The first one, CONV (normal), only includes standard convolutions, the second one, INV. CONV (offset), only includes amplitude/offset invariant convolutions, and the last one, INV. CONV (trend), only includes amplitude/linear trend invariant convolutions. These three models are only trained on the raw datasets and then tested on 5 different scenarios: (i) *no additional deformations*, (ii) *the addition of random offset*, sampled from uniform distribution between specific

ranges, as well as (iii) *the addition of random trend* with slope and intercept values sampled again with uniform probability, (iv) *combination of added random offset and trend* and (v) *combination of random offset shift and smooth random walk*. For the last deformation, the added synthetic trend is a random walk generated from a Gaussian distribution and smoothed by a rolling mean. In order to compare with learned invariances, we also include the prominent supervised contrastive learning method TS-TCC (Eldele et al., 2021). On each dataset, this model is pre-trained with contrastive losses leveraging the different deformation scenarios to capture invariance; then, it is fine-tuned for classification on the raw data. Its performances are also evaluated under the 5 different scenarios.

**Results.** Table 1 displays accuracy scores, which are organized by scenarios of increasing magnitude of deformations, starting from (i) *no additional deformations* and ending with (v) *offset shift and smooth random walk*, the closer one to trend deformation.

In the first scenario (normalized data), there is no deformation-related distribution shift between the train and the test sets. On average, CONV (normal) and TS-TCC perform slightly better than the hard-coded invariant networks INV. CONV (offset) and INV. CONV (trend). Outside of the *HandOutline* dataset, the performance difference between the two groups does not exceed 5.0 points of accuracy. This slight performance difference could be attributed to small class amplitude, offset, or linear trend dependencies. However, as soon as deformations are added (even small), CONV (normal) and TS-TCC performances drop significantly. In contrast, the performance of the deformation invariant network INV. CONV (trend) remains constant and becomes the best performer in most cases. On average, INV. CONV (trend) performs better than TS-TCC by 16.0 points of accuracy.

Regarding the specificities of each model, CONV (normal) is the most affected by deformations. Its performance drops by 48% when adding minor deformations (offset) and goes up to 59% with smooth random walk trends. This suggests that convolutions are highly sensitive to deformations, even small ones, commonly observed in time series. Appendix A.5.2 clearly illustrates the sensitivity of CONV (normal) by comparing the feature maps of its convolutional filters with those of INV. CONV (offset) and INV. CONV (trend) on the same time series. When the time series undergoes deformations, the feature map landscape of CONV changes drastically. In contrast, its invariant counterparts (INV. CONV) maintain a consistent structure according to their invariance. TS-TCC is the second most affected model. On average, its performance drops by at most 34%, suggesting that the learned invariance is only partial and does not generalize well on new observations. In contrast, convolutions with hard-coded invariance behave according to their properties. INV. CONV (offset) performance remains constant when offset deformations are added but decreases when more complex deformations are added. Interestingly, the performances are equivalent between the linear trend scenario and the offset + linear trend one, comforting the invariance property to offset. INV. CONV (trend) performance remains almost constant on scenarios including offset and linear trend deformations ((ii),(iii),(iv)). Note that the slight performance variations in these three scenarios are due to some padding effects. In the worst scenario, including a smooth random walk trend, the performance remains quite constant; it drops on average by 6% compared to 59% for CONV (normal) and 34% for TS-TCC. This last scenario is the closest to any trend deformation, indicating that INV. CONV (trend) is well suited for classifying time series affected by trends.

**Conclusion.** Standard convolutions are sensitive to spatiotemporal deformations. This sensitivity issue can be overcome by leveraging hard-coded invariant convolutions, which also benefit from better generalization properties than invariance learned by contrastive learning.

## 4.2. Classification Benchmark

**Pool of convolutions.** The choice of invariances is often related to the application (Yue et al., 2022), and setting the invariances by hand requires a good understanding of the nature of the signals. In the absence of such knowledge, as in classification benchmark, we decompose the space of deformations $H = \bigoplus_{i=1}^{K} H_i$ in the direct sum of subspaces such that the cumulative sums, $\emptyset \subset H_1 \subset H_1 + H_2 \subset \ldots \subset H$, represent sets of deformations of increasing order of complexity. We consider a layer that concatenates of $n_j$ ($\bigoplus_{i=1}^{j} H_i$)-invariant convolutions for $j \in (1, \ldots, K)$ and $n_0$ standard convolutions. In our specific experimental framework, the linear trend deformations $H_{LT}$ is the direct sum of offset deformations $H_1 = \{t \in I \mapsto b \mid b \in \mathbb{R}^D\}$ and purely linear deformations $H_2 = \{t \in I \mapsto at \mid a \in \mathbb{R}^D\}$. Illustrations of the proposed pool of convolutions is provided in the Figure 4 of Appendix A.2. Other pooling strategies, like attention mechanism or reinforcement learning, are possible and left for future work.

**Datasets.** We consider the 26 multivariate *UEA* datasets (Bagnall et al., 2018), coming with a standard train/test split. We also consider 3 additional datasets, the human activity recognition *UCIHAR* (Anguita et al., 2013) dataset, the *Sleep-EDF* dataset (Goldberger et al., 2000) for sleep stage classification of EEG signals, and finally the epileptic seizure recognition *Epilepsy* dataset (Andrzejak et al., 2001). For these datasets, we follow the same preprocessing with (Eldele et al., 2021), deriving train/validation/test sets of $60 : 20 : 20$ ratio. Appendix A.3 provides additional details on data splits and preprocessing.

**Baselines.** We select twelve state-of-the-art models for time series classification. TIMESNET (Wu et al., 2022), PATCHTST (Nie et al., 2022), CROSSFORMER (Zhang & Yan, 2022), DLINEAR (Zeng et al., 2023), iTRANSFORMER (Liu et al., 2023) and TIMEMIXER (Wang et al., 2024a) are derived from the Time-Series-Library (Wang et al., 2024b). Additionally, PERI-MIDFORMER (Wu et al., 2024) model is adapted from its official code repository. We also compare to the CNN-based backbone of the self-supervised method TSLANET (Eldele et al., 2024) and three powerful CNN architectures, namely INCEPTION (Ismail Fawaz et al., 2020), RESNET (Wang et al., 2017), and CNN (Ismail Fawaz et al., 2018) build upon standard stacked 1D-CNNs. We also include the state-of-the-art machine learning method ROCKET (Dempster et al., 2020). Finally, our model INVCONVNET combines a single inception-like pool of convolutions layer of four different kernel sizes, combining balanced sets of

Table 2: Classification Accuracy (%) for all considered datasets. Accuracy is averaged for datasets from *UEA* repository. Higher is better, best methods in **bold**, second best underlined.

| Datasets | INVCONVNET (ours) | TIMESNET (2022) | PATCHTST (2022) | CROSSFORMER (2022) | iTRANSF. (2023) | PERI-MID. (2024) | TSLANET (2024) | DLINEAR (2023) | TIMEMIXER (2024a) | INCEPTION (2020) | RESNET (2017) | CNN (2018) | ROCKET (2020) |
|---|---|---|---|---|---|---|---|---|---|---|---|---|---|
| *UEA (26 datasets)* | **71.81 ± 0.80** | 66.87 ± 1.72 | 66.18 ± 1.26 | 66.37 ± 1.35 | 64.42 ± 2.05 | 60.87 ± 3.00 | 68.70 ± 1.19 | 61.51 ± 1.05 | 64.60 ± 1.69 | 62.86 ± 1.96 | 67.37 ± 1.59 | 65.67 ± 1.64 | 71.29 ± 0.90 |
| *UCIHAR* | **96.63 ± 0.49** | 91.66 ± 0.62 | 85.74 ± 0.50 | 93.43 ± 0.56 | 94.13 ± 0.03 | 92.64 ± 0.57 | 94.71 ± 0.66 | 57.47 ± 0.73 | 84.40 ± 1.07 | 95.26 ± 0.55 | 96.04 ± 0.48 | 95.78 ± 0.20 | 92.06 ± 0.15 |
| *Sleep-EDF* | 84.95 ± 0.39 | 74.64 ± 0.73 | 78.53 ± 0.28 | 79.82 ± 0.89 | 53.11 ± 0.21 | 63.50 ± 1.52 | 84.98 ± 0.43 | 36.15 ± 0.21 | 74.07 ± 2.05 | 84.06 ± 0.39 | **85.62 ± 0.13** | 82.41 ± 0.56 | 83.88 ± 0.09 |
| *Epilepsy* | **98.43 ± 0.04** | 97.62 ± 0.20 | 98.01 ± 0.05 | 98.23 ± 0.12 | 97.80 ± 0.26 | 97.99 ± 0.29 | 98.23 ± 0.05 | 82.26 ± 0.06 | 97.94 ± 0.18 | 97.65 ± 0.20 | 98.16 ± 0.04 | 97.61 ± 0.28 | 98.38 ± 0.02 |

Table 3: Classification ablation study of filter types in INVCONVNET, including solely normal ones (-N), offset-invariant (-O) ones, or trend-invariant ones (-T).

| Datasets | INVCONVNET - mixed - | INVCONVNET-N - normal - | INVCONVNET-O - offset - | INVCONVNET-T - trend - |
|---|---|---|---|---|
| *UEA (26 datasets)* | **71.81 ± 0.80** | 68.04 ± 1.76 | 67.70 ± 1.21 | 66.61 ± 2.58 |
| *UCIHAR* | **96.63 ± 0.49** | 96.13 ± 0.34 | 95.75 ± 0.51 | 95.43 ± 0.13 |
| *Sleep-EDF* | **84.95 ± 0.39** | 84.70 ± 0.39 | 83.73 ± 0.13 | 83.14 ± 0.13 |
| *Epilepsy* | **98.43 ± 0.04** | 98.09 ± 0.13 | 98.26 ± 0.04 | 98.25 ± 0.09 |

standard, amplitude/offset and amplitude/linear trend invariants convolutions. A linear classifier follows the pool of convolutions. Implementation details are presented in Appendix A.2.

**Results.** Table 2 presents the classification performance of the proposed INVCONVNET against the twelve considered baselines, evaluated on *UEA* and the 3 additional datasets. All models are trained and tested for 3 runs with random seeds, and the average accuracy with its standard deviation is reported. Full classification results per dataset on *UEA*, along with a critical difference diagram based on mean ranks, are provided in Appendix A.5.3 (see also Figure 10). We observe that on *UEA* repository, INVCONVNET has the best average test classification accuracy, followed closely by ROCKET, and both algorithms outperform the rest deep learning approaches. ROCKET's use of thousands of random convolutional kernels enables effective feature extraction, making it highly robust on smaller *UEA* classification datasets. Although InvConvNet has the highest mean accuracy, ROCKET is the best method in first-rank counts, but in the critical difference diagram both show statistical equivalence in mean rank. Still, InvConvNet's top mean rank indicates more consistent performance, suggesting robustness across diverse datasets. On the additional 3 datasets, INVCONVNET shows superior performance in terms of accuracy, further validating the advantage of invariant CNN-based approaches in classification. For *Sleep-EDF*, RESNET is slightly better than the proposed INVCONVNET, which can be attributed to the depth of the method in capturing complex dependencies between the time series inputs. Finally, transformer-based methods are significantly outcompeted by CNN-based ones, and the MLP-based DLINEAR scores the worst, failing to capture the class-dependent temporal dynamics. Additionally, INVCONVNET built upon a single layer of convolutional kernels shows a significant

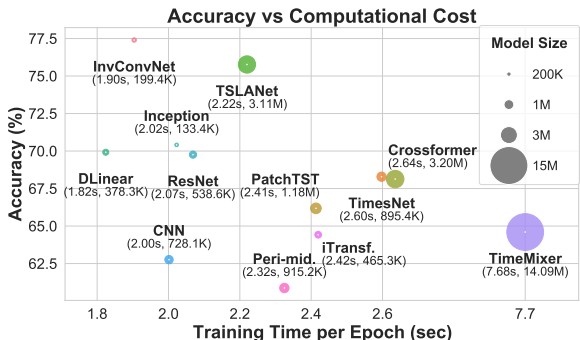

Figure 3: Models' costs for *Heartbeat* dataset classification.

advantage in terms of training time and memory cost, as presented in Figure 3 for the *UEA Heartbeat* dataset.

**Ablation study.** In this experiment, we compare INVCONVNET classification performances to its standalone main components. More specifically, INVCONVNET combines balanced sets of standard, amplitude/offset, and amplitude/linear trend invariant convolutions. We compare this model to its standards INVCONVNET-N, amplitude/offset INVCONVNET-O, and amplitude/linear trend INVCONVNET-T counterparts. In all cases, the total number of kernels remains the same. Table 3 depicts the classification performances of all four variants. INVCONVNET shows the highest performance on all datasets. INVCONVNET-N is second in most cases, closely followed by the invariant models. It indicates that features invariants to amplitude, offset, or linear trend are valuable on many classification tasks while guaranteeing robustness to the networks as seen previously in Section 4.1, and by combining them, models achieve better performances.

**Conclusion.** Convolutional features invariant to deformations are meaningful for classification tasks, and combining them in single-layer simple and lightweight architectures, offers comparable to better performances compared to the latest and prominent methods.

### 4.3. Transfer Learning Experiment.

**Protocol.** We assess the generalization of invariant convolutions on a transfer learning experiment using 4 different source and target domains of the *Fault-Diagnosis*

Table 4: Classification Accuracy (%) on a Transfer Learning experiment on *Fault-Diagnosis* (A, B, C, D sub-datasets) for the supervised INVCONVNET and INVCONVNET-N (normal) and two self-supervised methods.

| Methods | A→B | A→C | A→D | B→A | B→C | B→D | C→A | C→B | C→D | D→A | D→B | D→C | Avg. Acc. (%) |
|---|---|---|---|---|---|---|---|---|---|---|---|---|---|
| TS-TCC *(FT)* | 55.33 ± 1.44 | 52.52 ± 4.55 | **62.13 ± 1.39** | 48.05 ± 3.32 | 71.50 ± 1.83 | **100.0 ± 0.0** | 40.76 ± 2.22 | **98.25 ± 1.22** | **99.34 ± 0.50** | 46.98 ± 0.65 | **100.0 ± 0.0** | 74.28 ± 2.77 | 70.76 ± 1.66 |
| TS2VEC *(FT)* | 54.11 ± 1.46 | 54.07 ± 1.91 | 52.54 ± 1.89 | 55.06 ± 0.17 | **88.72 ± 0.47** | **100.0 ± 0.0** | 57.81 ± 2.18 | 78.30 ± 3.80 | 78.41 ± 4.39 | 60.37 ± 1.95 | 99.97 ± 0.02 | **86.82 ± 0.54** | 72.18 ± 1.57 |
| INVCONVNET *(Sup.)* | 55.90 ± 0.42 | **55.93 ± 0.34** | 53.41 ± 0.14 | **85.10 ± 0.63** | 78.54 ± 0.17 | 99.05 ± 0.08 | **70.75 ± 1.32** | 85.04 ± 0.13 | 85.12 ± 0.15 | **70.91 ± 0.73** | **100.0 ± 0.0** | 78.49 ± 0.38 | **76.52 ± 0.37** |
| INVCONVNET-N *(Sup.)* | **60.55 ± 0.88** | 55.50 ± 1.82 | 53.50 ± 0.85 | 60.26 ± 2.01 | 77.30 ± 0.60 | 93.50 ± 0.90 | 64.93 ± 0.48 | 84.87 ± 0.23 | 84.46 ± 0.51 | 59.98 ± 0.98 | 99.96 ± 0.0 | 77.14 ± 0.14 | 72.66 ± 0.78 |

dataset (Lessmeier et al., 2016). Model-wise, we include the self-supervised methods TS-TCC (Eldele et al., 2021) and TS2VEC (Yue et al., 2022) as well as INVCONVNET and its standard counterpart INVCONVNET-N configuration as presented in Section 4.2. Models are trained and tested on different source and target domains, namely the A, B, C, and D sub-datasets of *Fault-Diagnosis*, with direct transfer as in (Eldele et al., 2021). The self-supervised methods are pre-trained and fine-tuned (FT) on each source domain dataset leveraging contrastive learning.

**Results.** Table 4 presents the transferred classification accuracy performances. Interestingly, the supervised IN-vCONVNET, built on invariant convolutions, outperforms unsupervised methods and its standard counterpart (-N) by at least 4% with low variance.

**Conclusion.** This experiment indicates that transfer learning tasks on time series can benefit from convolutional invariant features to remove the distribution shift caused by deformations and improve generalization.

## 5. Conclusion & Future Work

In this study, we establish a formal mathematical framework for time series invariances, and we design exact hard-coded invariant convolutions that seamlessly integrate within any CNN-based model. We experimentally show their enhanced generalizability and computational efficiency in several setups. To showcase the robustness of our framework, we focus our invariant convolutions to simpler but common affine deformations, allowing invariance to trend when it can be assumed linear at the scale of the convolution kernel size. However, we plan to explore approximating the trend with non-linear functions, such as splines or higher-degree polynomials, and incorporating seasonal components with low-frequency cosine bases. Finally, exploring additional merging strategies on different types of invariances and extending our invariant convolutional layers to unsupervised learning settings, is on our agenda for future work.

## Acknowledgements

This work has been partially funded by the Industrial Data Analytics and Machine Learning (IdAML) chair hosted at ENS Paris-Saclay, Université Paris-Saclay.

## Impact Statement

The work presented in this paper aims to advance the field of Machine Learning for time series analysis by introducing a novel architectural design of convolutional layers that are invariant to specific group actions for time series data, built upon a thorough mathematical framework. Based on the presented experimental evaluation, the proposed invariant convolutional layers have the potential to significantly advance diverse domains of time series modeling, such as healthcare, environmental monitoring, and industrial machinery. By enabling robust time series representations and insights into their invariant properties, these layers can address critical challenges in real-world applications and tasks, including classification and anomaly detection, among others.

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

# A. Appendix

## A.1. Invariant embedding

Let M be a Hilbert space and H a finite dimensional vector subspace of M. We focus on the action H on M by the usual vector addition: $(h, m) \in H \times M \mapsto m + h \in M$. The following proposition exhibits a H-invariant embedding that is also orbit-injective.

*Proposition* 3. Let $P_H$ be the orthogonal projector on H, and $I_d$ be the identity map on M, the embedding, $L = I_d - P_H$ (the projector on $H^\perp$) is H-invariant and orbit-injective.

*Proof.* **Existence of** $L$: As H is a finite dimension vector space, it is a closed and convex subset of the Hilbert space M; the orthogonal projector on H, denoted $P_H$, exists. Therefore, $L : m \in M \mapsto m - P_H(m) \in H$ is well defined.

H-**invariance of** $L$: Since H is closed, $M = H \oplus H^\perp$, and for any $x \in M$, we decompose $m = m_H + f_{H\perp}$. Thus, for any $m \in M$, and $h \in H$:

$$\begin{aligned}
L(m + h) &= m + h - P_H(m + h) \\
&= m + h - P_H(m_{H\perp} + m_H + h) \\
&= m + h - (m_H + h) \quad \textbf{(projector on a closed vectorial subspace)} \\
&= m - m_H \\
&= L(m)
\end{aligned}$$

which proves the H-invariance of $L$.

**Orbit-injectivity of** $L$: For any $m \in M$, its orbits corresponds to:

$$\begin{aligned}
[m] &= \{m + h \mid h \in H\} \\
&= \{L(m) + h' \mid h \in H, \ h' = P_H(m) + h \in M\} \\
&= L(m) + H
\end{aligned}$$

Therefore, for any $([m], [m']) \in M/H \times M/H$, such that $[m] \cap [m'] = \emptyset$ implies that $L(m) \neq L(m')$ proving the orbit-injectivity of $L$. $\qquad\square$

## A.2. INVCONVNET: Architectural Details

The proposed pool of convolutions is presented in Figure 4 and is built upon the concatenation of normal filters, offset shift-invariant filters, and linear trend invariant filters. The proposed convolutional layer, therefore, incorporates three distinct kernel types with respect to invariance.

### A.2.1. INVARIANT EMBEDDING MODULES

After mathematically formulating the characteristics of an invariant convolutional layer, which is built upon a standard (or variant) kernel, a kernel invariant to offset shift and scaling, and a kernel invariant to linear trend and scaling, we provide additional details for the design of the employed embedding modules. We present visualizations of the embedding modules used for classification and anomaly detection (i. e., reconstruction) in Figure 5.

**Standard Module (Single-Layer):** The simplest embedding module is a single invariant convolutional layer for a specific kernel size $W$ and hidden dimensions $d_{n_0}$ for the standard convolutional part (in yellow), $d_{n_1}$ for the convolutional part invariant to offset shift and scaling (in red), and $d_{n_2}$ for the convolutional part invariant to linear trend and scaling (in purple).

**Inception-like Module (Single-Layer):** We also study inception-like design by employing several kernel sizes, but without stacking the layers in increasing depths. The depth of the employed module remains equal to one. As shown in Figure 5 (Left), in an inception-like embedding module, we consider several kernel sizes, e. g., $W_1, W_2, W_3$, that are applied in parallel to the input series, while leveraging the three parts of the proposed pool of convolutions (including standard and invariant ones). The produced representation for the different kernel sizes, i. e., $\mathbf{z_1}, \mathbf{z_2}, \mathbf{z_3}$ are concatenated in the channel dimension producing embeddings of size $(3 * \sum_{j=0}^{2} d_{n_j}, L)$ for 3 selected kernel sizes. The distinct kernel sizes as well as the hidden dimension for each part in the pool of convolutions, are hyperparameters that we need to tune, as in every CNN-based architecture.

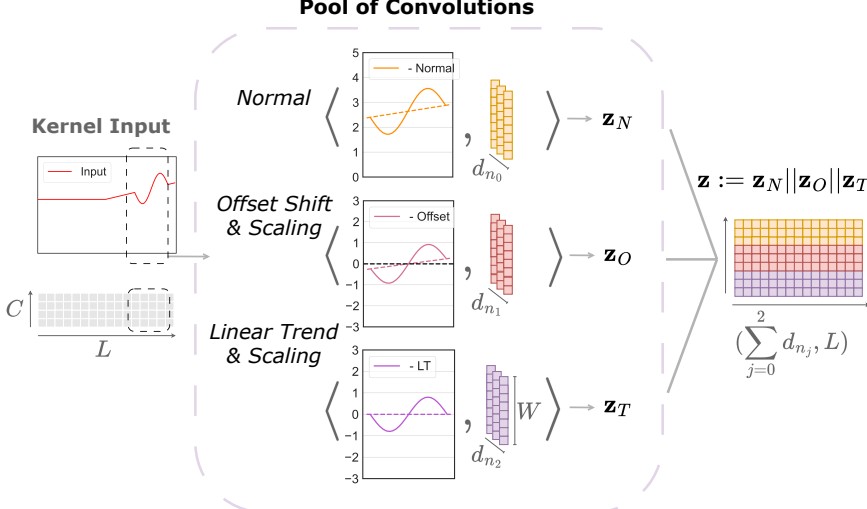

Figure 4: Visualization of the different kernel types employed on an input signal inside the proposed invariant convolutional layer, including normal filters (in yellow), filters invariant to offset shift (in red) and filters invariant to linear trend (in purple). The produced embedding $\mathbf{z}$ is the result of concatenations of the different representations.

**Multi-Scale Module (Multi-Layer):** Additionally, we examine the capacity of a multi-scale embedding module built upon invariant convolutions as presented in Figure 5 (Right), particularly for the reconstruction task. Here the employed depth is equal to two. At the first level, an inception-like layer is employed, with kernel sizes selected to be powers of two (deriving the maximum exponent from the logarithm of half the series length and setting the minimum to four). The kernels, as described above, are applied in parallel, and the produced representation for the different kernel sizes is concatenated in the channel dimension. At the second level, a standard convolutional layer is applied. We similarly employ several kernel sizes of size $\max(W_i)/W_i$ matching the picked kernel sizes in the first layer $W_i$ for $i \in \{1, \ldots, K\}$, where $K$ the number of kernels with distinct sizes. For each distinct kernel size in this second layer, a dilation factor is set as $r_i = W_i$, thus equal to the kernel size of the previous layer. This design enables capturing representations at different scales while employing a shallow and computationally light architecture, that still benefits from invariant convolutions. Experimentally, this module shows performance improvements for the anomaly detection task, where reconstruction can benefit from capturing dependencies at different granularities.

### A.2.2. TASK-SPECIFIC MODULES

The embeddings derived from the modules described above are further processed by standard layers to produce the task-specific output.

**I. Classification.** The embedding is passed by a Global Average Pooling (GAP) layer, applied to the channel dimension, that averages over the time dimension to reduce the temporal features to a single value per channel. The result of the GAP layer is followed by a linear layer that produces the final class probabilities.

**II. Anomaly Detection (Reconstruction).** The multi-layer embedding module presented above is used to capture dependencies at various scales. For the reconstruction of the input, the embedding is followed by linear layers applied first on the temporal and then on the channel dimensions. The coefficients of the invariant kernels, i.e., $(\sigma_f, \mu_f)$ for the offset-invariant part and $(\nu_f, \alpha_f, \beta_f)$ for the trend-invariant part, are passed through linear layers to be mapped to the original time and channel dimensions and are then combined with each part (one variant and two invariant) of the embedding via addition and multiplication. More specifically, $\sigma_f$ and $\nu_f$ refer to the norm of invariant embeddings of the signal, $\mu_f$ refers to the means, $\alpha_f$ and $\beta_f$ refers to the coefficients of the linear trend. Combining the coefficients with the linearly projected embeddings adapts the level of the series to the original one, enabling enhanced temporal resolution. Figure 6 presents details about the reconstruction module and its relation with the embedding module.

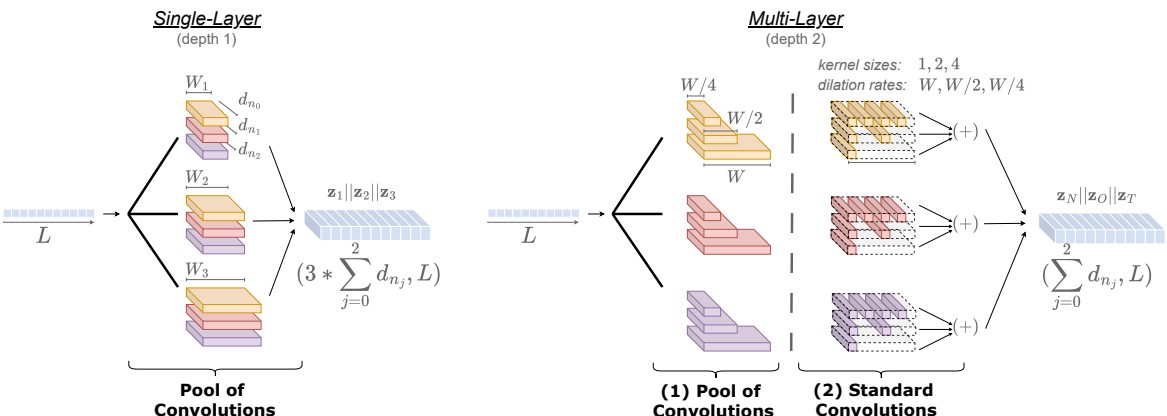

Figure 5: **Left:** Single-layer embedding module, i. e., of depth 1, used in standard INVCONVNET (for 1 chosen kernel width $W$) and in inception-like INVCONVNET (for several chosen kernel widths, e. g., 3 visualized in the figure). **Right:** Multi-layer embedding module, i. e., of depth 2, used in multi-scale INVCONVNET (for several chosen kernel widths $W$). In the second layer, for each kernel size, we utilize multiple kernels whose total number sums up to the larger kernel size, achieving multi-scale views.

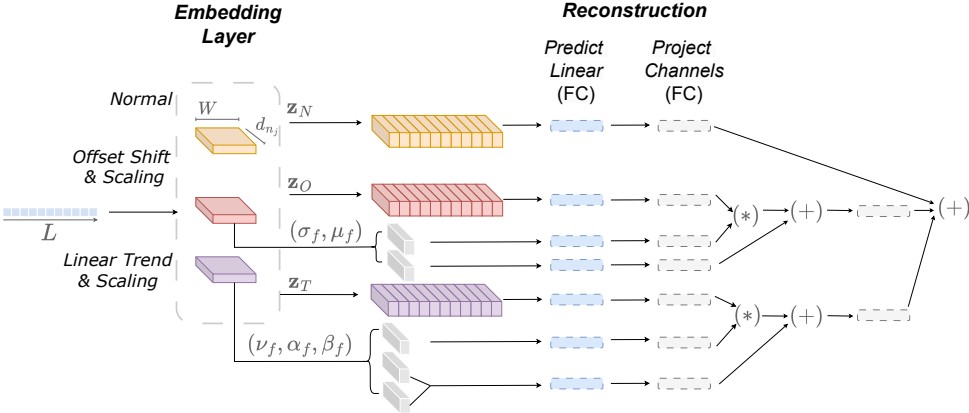

Figure 6: Visualization of the architecture used in terms of reconstruction that leverages the output of an embedding module built upon a pool of convolutions (including variant and invariant ones). The representation of the embedding layer $\mathbf{z}_N, \mathbf{z}_O, \mathbf{z}_T$ are passed from 2 fully connected linear layers (FC), from which the first operates on the temporal dimension and the second on the channels. The coefficients of each invariant operation are similarly projected with linear layers and combined with the representation (with addition or multiplication) to produce the output.

## A.3. Datasets Details

We focus our experimental evaluation on several real-world time series datasets, including univariate and multivariate inputs, with significant applications in healthcare and medical diagnosis, wearable technology, audio processing, and transportation, among others.

**Classification Datasets.** Details about the 26 multivariate derived from *UEA* data repository (Bagnall et al., 2018), that are employed in terms of this study in the classification experiment are provided in Table 5. More specifically, for each dataset we mention the number of channels, the length of the multivariate series, as well as the number of classes and the number of instances in the predefined train and test sets.

Table 5: Details of *UEA* datasets used for classification.

| Dataset | #Train | #Test | #Channels | Length | #Classes |
|---|---|---|---|---|---|
| *ArticularyWordRecognition* | 275 | 300 | 9 | 144 | 25 |
| *AtrialFibrillation* | 15 | 15 | 2 | 640 | 3 |
| *BasicMotions* | 40 | 40 | 6 | 100 | 4 |
| *Cricket* | 108 | 72 | 6 | 1197 | 12 |
| *Epilepsy* | 137 | 138 | 3 | 206 | 4 |
| *EthanolConcentration* | 261 | 263 | 3 | 1751 | 4 |
| *FaceDetection* | 5890 | 3524 | 144 | 62 | 2 |
| *FingerMovements* | 316 | 100 | 28 | 50 | 2 |
| *HandMovementDirection* | 320 | 147 | 10 | 400 | 4 |
| *Handwriting* | 150 | 850 | 3 | 152 | 26 |
| *Heartbeat* | 204 | 205 | 61 | 405 | 2 |
| *InsectWingbeat* | 30000 | 20000 | 200 | 78 | 10 |
| *JapaneseVowels* | 270 | 370 | 12 | 29 | 9 |
| *Libras* | 180 | 180 | 24 | 51 | 6 |
| *LSST* | 2459 | 2466 | 6 | 36 | 14 |
| *MotorImagery* | 278 | 100 | 64 | 3000 | 2 |
| *NATOPS* | 180 | 180 | 24 | 51 | 6 |
| *PEMS-SF* | 267 | 173 | 963 | 144 | 7 |
| *PenDigits* | 7494 | 3498 | 2 | 8 | 10 |
| *PhonemeSpectra* | 3315 | 3353 | 11 | 217 | 39 |
| *RacketSports* | 151 | 152 | 6 | 30 | 4 |
| *SelfRegulationSCP1* | 268 | 293 | 6 | 896 | 2 |
| *SelfRegulationSCP2* | 200 | 180 | 7 | 1152 | 2 |
| *SpokenArabicDigits* | 6599 | 2199 | 13 | 93 | 10 |
| *StandWalkJump* | 12 | 15 | 4 | 2500 | 3 |
| *UWaveGestureLibrary* | 120 | 320 | 3 | 315 | 8 |

Table 6: Details of additional datasets used for classification.

| Dataset | #Train | #Test | #Channels | Length | #Classes |
|---|---|---|---|---|---|
| *UCIHAR* | 7352 | 2947 | 9 | 128 | 6 |
| *Sleep-EDF* | 25612 | 8910 | 1 | 3000 | 5 |
| *Epilepsy* | 9200 | 2300 | 1 | 178 | 2 |
| *Fault-Diagnosis* | 8184 | 2728 | 1 | 5120 | 3 |

Table 6 contains the same details for the additional datasets used for classification, i.e., the *UCIHAR* (Anguita et al., 2013) dataset, the *Sleep-EDF* dataset (Goldberger et al., 2000), and the *Epilepsy* dataset (Andrzejak et al., 2001). More specifically, *UCIHAR* data were collected by 30 volunteers performing various activities, including laying, standing, sitting, walking, walking downstairs, and walking upstairs. Volunteers' records were captured by a waist smartphone, including distinct measurements connected to acceleration and velocity signals. The *Sleep-EDF* dataset from the PhysioBank database consists of PolySomnoGraphic sleep recordings containing EEG, among other measurements. We consider only the EEG signals following previous studies (Eldele et al., 2021) and performed sleep stage classification, including awake, rapid eye movement, and non-rapid eye movements. Finally, the *Epilepsy* dataset consists of EEG brain activity measurements for epileptic seizure classification. Following the preprocessing of (Eldele et al., 2021), we perform binary classification after merging classes referring to non-epileptic seizure. For the transfer learning experiment, we utilized the *Fault-Diagnosis*

dataset (Lessmeier et al., 2016), as preprocessed in (Eldele et al., 2021), which comprises of measurements under 4 different working conditions, perceived as different domains, and are assigned to 3 classes, including a healthy and two fault classes.

We also conduct a synthetic experiment on 5 large datasets from *UCR* repository (Dau et al., 2019). The selected datasets were picked from the equal-length datasets of the relevant repository that combine a large number of samples in the training and test sets, along with a large series length. Datasets with train set size and series length less than 100 were not considered in our subset. Details about the selected *UCR* datasets are given in Table 7. Since many *UCR* datasets are already preprocessed using Z-normalization to achieve zero mean and unit variance, they are not ideal for demonstrating the impact of our invariant layers on classification performance. This is the primary reason for conducting a synthetic experiment on these datasets rather than a conventional one with the whole repository.

Table 7: Details of the subset of *UCR* datasets used for the synthetic classification experiment.

| Dataset | #Train | #Test | #Channels | Length | #Classes |
|---|---|---|---|---|---|
| *HandOutlines* | 1000 | 370 | 1 | 2709 | 2 |
| *MixedShapesRegularTrain* | 500 | 2425 | 1 | 1024 | 5 |
| *NonInvasiveFetalECGThorax1* | 1800 | 1965 | 1 | 750 | 42 |
| *FordB* | 3636 | 810 | 1 | 500 | 2 |
| *Yoga* | 300 | 3000 | 1 | 426 | 2 |

**Anomaly Detection Datasets.** Furthermore, we present in Table 8 the five employed anomaly detection datasets after preprocessing them on non-overlapping subsequences of length 100, also showing the number of channels and the size of the train, validation, and test splits. The *SMD* dataset (Su et al., 2019) consists of data related to server machines collected at an internet company, while the *MSL* and *SMAP* (Hundman et al., 2018) datasets comprise of telemetry data from spacecraft monitoring systems. The *SWaT* (Mathur & Tippenhauer, 2016) dataset is a collection of sensor data from the operations of a critical infrastructure system. Finally, the *PSM* (Abdulaal et al., 2021) dataset contains measurements from application server nodes on an internet website.

Table 8: Details of datasets used for anomaly detection.

| Dataset | #Train | #Val | #Test | #Channels | Length |
|---|---|---|---|---|---|
| *SMD* | 566724 | 141681 | 708420 | 38 | 100 |
| *MSL* | 44653 | 11664 | 73729 | 55 | 100 |
| *SMAP* | 108146 | 27037 | 427617 | 25 | 100 |
| *SWaT* | 396000 | 99000 | 449919 | 51 | 100 |
| *PSM* | 105984 | 26497 | 87841 | 25 | 100 |

**Data Splits and Preprocessing.** As mentioned already in the main paper, for the proposed method, we do not normalize the data using Z-normalization for *UEA* and the rest 4 datasets used in classification, while the datasets from *UCR* are used for the synthetic experiment are derived normalized by the data source. On the contrary, all data are normalized for classification and the baselines, as well as for anomaly detection and all considered models (including the proposed INVCONVNET). For the *UEA* datasets, we do validation on the whole training set since the test sets are, in several cases, quite large, and thus, a small subset of the train set picked for validation can be a misleading indicator of performance. For the rest classification datasets, we perform a split into train/validation/test sets with a 60 : 20 : 20 ratio, following (Eldele et al., 2021). Similarly, for the five anomaly detection datasets, we split into train/validation/test sets with a 70 : 10 : 20 ratio (Xu, 2021).

### A.4. Implementation Details

All experiments presented in this study were conducted on an Nvidia Tesla V100 GPU, with 40 cores and 756 GB of memory. We utilized the Adam optimizer with a learning rate of lr = 0.001 for both classification and unsupervised anomaly detection tasks. We also adopted a linear cosine annealing learning rate scheduler for INVCONVNET in classification. More specifically, the scheduler started the warmup phase with a learning rate equal to 0.001, linearly increasing the learning rate over the first 10 epochs to 0.01. After the warmup, it gradually reduced the learning rate using a cosine annealing schedule, down to 0.0001 by the end of training. For anomaly detection and the rest methods, we utilized a learning rate scheduler of 0.5 decrease rate per epoch. To have better estimates for the generalization performance of all models and, most importantly, our proposed shallow modules, we performed 3 runs with random seeds for all considered datasets and tasks. Additional

details for each task and the hyperparameters of the models are given below.

**- Classification Task:** We trained the models for 100 epochs for all *UEA* datasets, the 5 *UCR* datasets and the *Fault-Diagnosis* dataset. We performed early stopping during training, after 20 epochs of no improvement in the validation accuracy for all models and kept the configuration of weights that correspond to the best validation accuracy during training. The standard cross entropy loss was optimized during training for classification. For the INVCONVNET model, we considered the inception-like embedding module of Figure 5 (Left) for all datasets of *UEA* except for *Epilepsy*, *EthanolConcentration*, *Heartbeat* that we selected the standard embedding module of single kernel size. Finally, for *FingerMovements*, *Handwriting*, *Libras* and *SelfRegulationSCP1* datasets, we employed the multi-scale embedding layer of Figure 5 (Right). The type of the embedding layer, as well as the hyperparameters for the convolutional layers, e. g., kernel size and hidden dimension, were selected through random search and the best performance on the validation set. For the rest of the classification datasets, i. e., the *UCIHAR*, the *Sleep-EDF* and the *Epilepsy* datasets, we trained all models for 300 epochs with 20 epochs patience and considered the inception-like embedding layer, since it was performing better on the validation set.

**- Unsupervised Anomaly Detection Task:** We trained the models for 10 epochs and stopped training if no improvements had been made in terms of validation loss for 3 epochs, saving the best model weights on the validation set. We optimized the models using the mean squared error (MSE) between the real input sequences and the reconstructed ones. For all five anomaly detection datasets, we used the multi-scale embedding layer of Figure 5 (Right), followed by the reconstruction module built upon linear layers in Figure 6.

**- Hyperparameter Selection:** We next provide more information about the selection of the kernel sizes and hidden dimensions for the different embedding modules tested in terms of INVCONVNET. For the standard pool of convolutions with one specific kernel size $W$, we chose the kernel size as the minimum value between the value 50 and half of the length of the time series. For the inception-like embedding module, we selected several kernel sizes, such as 51,75,101, and 125, or factors of those values for which the length of the series is proportional. Finally, for the first layer (pool of convolutions) of the multi-scale module, we computed the kernel sizes as powers of two, starting from 16 up to a maximum of 128, based on the logarithmic scaling of half the series length. For all modules, we tested hidden dimensions sizes for the pool of convolutions in $\{32, 64, 128, 256\}$ doing a split that enabled almost equal contribution for the three parts, i. e., normal, invariant to offset shift and scaling, and invariant to linear trend and scaling. For instance for total hidden size equal to 32 the different parts had $(12, 10, 10)$ hidden dimensions respectively, for 64 the split became $(24, 20, 20)$ and so on.

For the common CNN-based baselines INCEPTION, RESNET, CNN, we tuned the number of convolutional layers, the kernel sizes, and the hidden size of each layer. We followed a random search for a value between 2 and 6 for the number of blocks and $\{32, 64, 128, 256\}$ for the hidden dimensions, whereas for the kernel sizes, we used those proposed in the relevant papers (Ismail Fawaz et al., 2020; Wang et al., 2017; Ismail Fawaz et al., 2018). All baselines' implementations are derived from the Time-Series-Library (Wang et al., 2024b), with the configurations mentioned in the respective papers, and the main code resources for performing the different tasks, e. g., classification and anomaly detection were adopted. We also used ROCKET (Dempster et al., 2020) from sktime Library (Löning et al., 2019), with 3000 random convolutional kernels. Finally, for the transfer learning classification experiment, the self-supervised contrastive TS-TCC and TS2VEC methods were trained with their default parameters for classification as proposed in the respective papers (Eldele et al., 2021; Yue et al., 2022), for 50 epochs for each phase of pre-training and fine-tuning.

### A.5. Additional Results

#### A.5.1. RECONSTRUCTION-BASED ANOMALY DETECTION BENCHMARK

Reconstruction-based anomaly detection involves training a model to learn a compact representation of normal data by reconstructing the input. The reconstruction error acts as the anomaly criterion, indicating whether the time series does not conform to the normal patterns based on a chosen threshold.

**Datasets.** For unsupervised anomaly detection, we deploy the following benchmark datasets; *SMD* (Su et al., 2019), *MSL* and *SMAP* (Hundman et al., 2018), *SWaT* (Mathur & Tippenhauer, 2016) and *PSM* (Abdulaal et al., 2021). We apply standard preprocessing to extract non-overlapping sub-sequences and split them into train/validation/test sets with a $70 : 10 : 20$ ratio (Xu, 2021; Wu et al., 2022).

**Baselines.** Focusing on reconstruction, we include fourteen time series (regression) models, most from Time-Series-Library (Wang et al., 2024b), including TIMESNET (Wu et al., 2022), PATCHTST (Nie et al., 2022), ETSFORMER (Woo

et al., 2022b), FEDFORMER (Zhou et al., 2022), AUTOFORMER (Wu et al., 2021), PYRAFORMER (Liu et al., 2021a), INFORMER (Zhou et al., 2021), REFORMER (Kitaev et al., 2020), LIGHTTS (Zhang et al., 2022), DLINEAR (Zeng et al., 2023), iTRANSFORMER (Liu et al., 2023), TIMEMIXER (Wang et al., 2024a), PERI-MIDFORMER (Wu et al., 2024) and the TSLANET (Eldele et al., 2024) backbone. For the decoder, we capitalize on the learned slope and intercept values for each of the two invariant embedding parts, i. e., the Offset and Linear Trend, to adjust the projected embedding back to the original temporal dimensions along with a linear layer. The projection to the initial channel dimension is then obtained by applying a second channel-wise linear layer. Details on the INVCONVNET embedding and reconstruction modules are also given in Appendix A.2.

**Results.** Table 9 shows F1-scores (%) for the proposed model and baselines across 5 anomaly detection datasets. IN-VCONVNET performs best on the *SWaT* dataset and ranks third in average performance across all datasets, slightly behind TIMESNET. The latter's superior performance can be attributed to its refined CNN blocks, which capture multiple periodicities for finer granularity in reconstruction. TSLANET achieves the highest average F1-score, benefiting from Fourier blocks before CNN modules to capture both short- and long-term dependencies. Other models, like the MLP-based DLINEAR and transformer-based FEDFORMER, also show competitive results. Notably, INVCONVNET excels with a single layer of invariant convolutions, proving the effectiveness of leveraging multi-scale invariances in shallow architectures.

Table 9: Anomaly Detection results in terms of the F1-score (%) for all considered datasets. Higher is better, best methods in **bold**, second best underlined.

| Datasets | INVCONVNET (ours) | TIMESNET (2022) | PATCHTST (2022) | TSLANET (2024) | ETSFORMER (2022b) | FEDFORMER (2022) | iTRANSF. (2023) | PERI-MID. (2024) | LIGHTTS (2022) | DLINEAR (2023) | TIMEMIXER (2024a) | AUTOFORMER (2021) | PYRAFORMER (2021a) | INFORMER (2021) | REFORMER (2020) |
|---|---|---|---|---|---|---|---|---|---|---|---|---|---|---|---|
| SMD | 84.05 ± 0.16 | **84.61 ± 0.56** | 84.15 ± 0.48 | 84.33 ± 0.17 | 79.69 ± 0.69 | 71.11 ± 0.02 | 82.38 ± 0.99 | 83.34 ± 0.63 | 83.04 ± 0.49 | 83.56 ± 0.14 | 83.20 ± 0.06 | 71.16 ± 0.02 | 71.36 ± 0.01 | 71.17 ± 0.03 | 71.22 ± 0.01 |
| MSL | 80.68 ± 0.01 | 80.33 ± 0.79 | 78.67 ± 0.04 | 74.65 ± 0.78 | 75.98 ± 0.54 | **82.06 ± 0.14** | 72.66 ± 0.04 | 80.93 ± 0.06 | 80.39 ± 0.06 | 81.92 ± 0.01 | 67.12 ± 3.33 | **82.08 ± 0.04** | 81.00 ± 0.08 | 82.02 ± 0.11 | 81.52 ± 0.08 |
| SMAP | 68.29 ± 0.07 | 69.18 ± 0.21 | 68.84 ± 0.01 | **80.26 ± 0.05** | 67.45 ± 0.74 | 68.71 ± 0.01 | 66.86 ± 0.08 | 67.62 ± 0.02 | 67.47 ± 0.02 | 67.32 ± 0.01 | 65.55 ± 0.32 | 75.28 ± 1.64 | 67.76 ± 0.13 | 68.74 ± 0.12 | 73.30 ± 0.15 |
| SWaT | **92.82 ± 0.19** | 92.71 ± 0.04 | 88.38 ± 1.11 | 91.65 ± 0.27 | 92.67 ± 0.06 | 79.18 ± 0.01 | 92.68 ± 0.01 | 92.17 ± 0.05 | 92.75 ± 0.01 | 92.66 ± 0.01 | 91.77 ± 1.30 | 79.18 ± 0.01 | 80.91 ± 0.38 | 79.75 ± 0.74 | 79.17 ± 0.01 |
| PSM | 96.34 ± 0.01 | **96.85 ± 0.27** | 96.12 ± 0.01 | 96.20 ± 0.03 | 95.23 ± 0.03 | 89.44 ± 0.88 | 95.15 ± 0.14 | 96.31 ± 0.09 | 95.50 ± 0.02 | 96.66 ± 0.01 | 94.01 ± 0.77 | 88.25 ± 0.01 | 93.66 ± 0.13 | 90.55 ± 0.05 | 90.74 ± 0.09 |
| **Avg. F1 (%)** | 84.44 ± 0.09 | 84.74 ± 0.37 | 83.23 ± 0.33 | **85.42 ± 0.26** | 82.20 ± 0.41 | 78.10 ± 0.21 | 81.95 ± 0.25 | 84.07 ± 0.17 | 83.83 ± 0.12 | 84.42 ± 0.04 | 80.33 ± 1.16 | 79.19 ± 0.65 | 78.94 ± 0.15 | 78.45 ± 0.21 | 79.19 ± 0.07 |

We also evaluate the 15 models on the 5 anomaly detection datasets using mean ranks based on F1-score, as shown in the critical difference diagram in Figure 7. Statistical significance was assessed using the Friedman test, as implemented in the Aeon Library (Middlehurst et al., 2024). Since the Friedman test did not indicate a significant difference at the $\alpha = 0.1$ level, no pairwise tests were applied, and all models fall within the same group, which can be attributed to the small number of considered datasets. Interestingly, in terms of mean rank scores, INVCONVNET ranks second with 4.8, closely following TIMESNET, which achieved the best mean rank of 3.8, demonstrating robust performance.

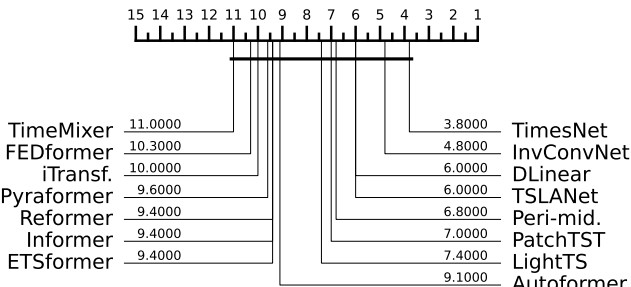

Figure 7: Critical difference diagram based on mean ranks from F1 scores on the 5 anomaly detection datasets.

Finally, in Table 10, we perform additional comparisons, in terms of anomaly detection, including the INVCONVNET model and the standard CNN-based variants, namely INCEPTION, RESNET and CNN originally proposed for classification. All models have an identical reconstruction module, with the exception of INVCONVNET, which also includes the signal decomposition coefficients on the invariant basis.

As observed, our proposed method consistently outperforms all evaluated CNN-based variants, suggesting the effectiveness of combining invariances with their related signal coefficients for reconstruction.

**Conclusion.** Invariant convolutional features combined with their related coefficients appear to be a concise and suited representation of time series for unsupervised anomaly detection based on reconstruction.

Table 10: Anomaly Detection results for INVCONVNET and vanilla CNN-based methods. Performance mentioned in terms of the F1-score (%). Higher is better, best methods in **bold**, second best underlined.

| Datasets | INVCONVNET | INCEPTION | RESNET | CNN |
|---|---|---|---|---|
| | | + *predict linear, project channels* | | |
| SMD | **84.05 ± 0.16** | 71.48 ± 0.11 | 76.20 ± 0.52 | 77.31 ± 0.91 |
| MSL | 80.68 ± 0.01 | **81.68 ± 0.08** | 81.25 ± 0.09 | 79.96 ± 0.20 |
| SMAP | 68.29 ± 0.07 | **68.63 ± 0.16** | 67.24 ± 0.65 | 67.00 ± 0.07 |
| SWaT | **92.82 ± 0.19** | 82.69 ± 0.70 | 80.93 ± 0.04 | 80.24 ± 0.95 |
| PSM | **96.34 ± 0.01** | 92.02 ± 0.35 | 92.30 ± 0.82 | 93.30 ± 0.58 |
| **Avg. F1 (%)** | **84.44 ± 0.09** | 79.30 ± 0.28 | 79.58 ± 0.42 | 79.56 ± 0.54 |

### A.5.2. ROBUSTNESS STUDY - VISUALIZATION OF FEATURE MAPS

In Figure 8, we provide visualizations of the feature maps produced from different filter types (among the ones introduced in the paper), which are later incorporated in INVCONVNET example architecture, including normal ones (CONVNET (normal)), filters invariant to offset shift (INVCONVNET (offset)) and filters invariant to linear trend (INVCONVNET (trend)). Specifically, we consider *FordB* dataset from the large *UCR* datasets, considered in the Robustness Study of Table 1.

For the considered dataset, the convolutional filters presented below are trained on normalized raw data and tested on four additional synthetic deformation scenarios: (i) *random offset (off.)*, (ii) *random linear trend (LT)*, (iii) *combined offset and trend (off., LT)*, and (iv) *combined offset and smooth random walk (off., RW)*. For the last deformation, the added synthetic trend is a random walk generated from a Gaussian distribution and smoothed by a rolling mean.

**Extraction of Feature Maps.** After passing the input series through the convolutional layer of each model and the activation function (i. e., ReLU(.)), we extract the feature map for each filter (or each hidden dimension) corresponding to the largest considered kernel size. We recall that for the synthetic experiment on the 4 larger *UCR* datasets, we leveraged the inception-like embedding module of Figure 5 (Left), built upon 4 different kernel sizes with each having an equal total number of filters (or hidden dimensions equal to 128). Please note that averaging over all outputs produced by the layer for the several distinct kernel sizes produces multi-scale representations that produce similar (in terms of activated regions) but smoother maps (in terms of intensity values). The resulting feature maps, which are derived by the activated outputs for the largest kernel size, are essentially 2D representations with dimensions equal to the series length $L$ and the 128 hidden dimensions.

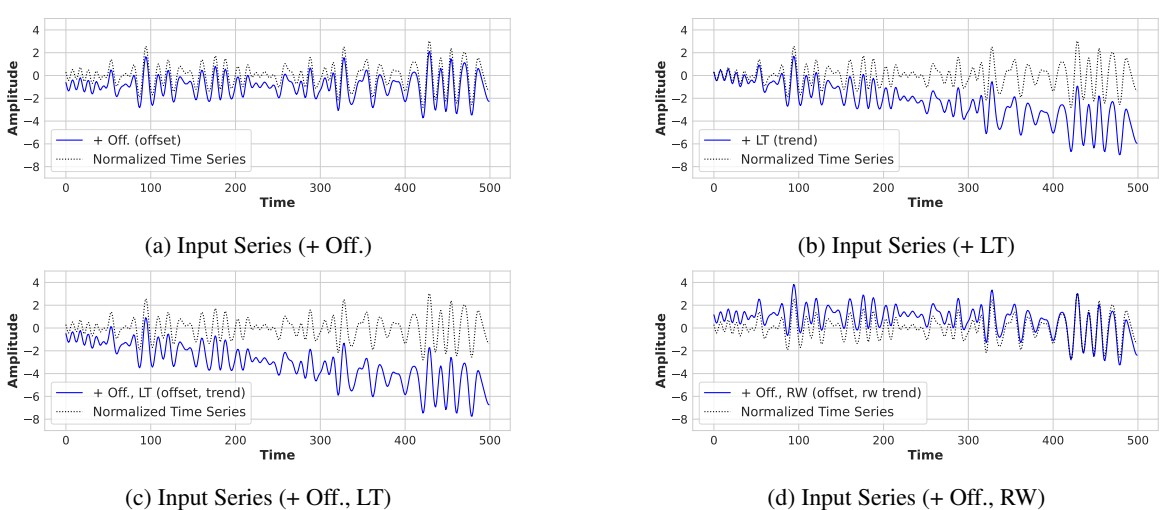

(a) Input Series (+ Off.)  (b) Input Series (+ LT)

(c) Input Series (+ Off., LT)  (d) Input Series (+ Off., RW)

Figure 8: Plot of a single sample from the test set of *FordB* dataset for *UCR* used to produce the example feature maps of different types of convolutional filters. The normalized time series is captured in a black dotted line, while signals in blue represent the deformed versions of the series. Specifically, (a) represents the sample with the addition of random offset, (b) represents the sample with the addition of random linear trend, (c) represents the sample with the addition of random offset and linear trend, and (d) represents the sample with the addition of random offset and random walk trend.

We provide in Figure 9 the activated feature maps for a single sample for *FordB* dataset and its deformed versions (as

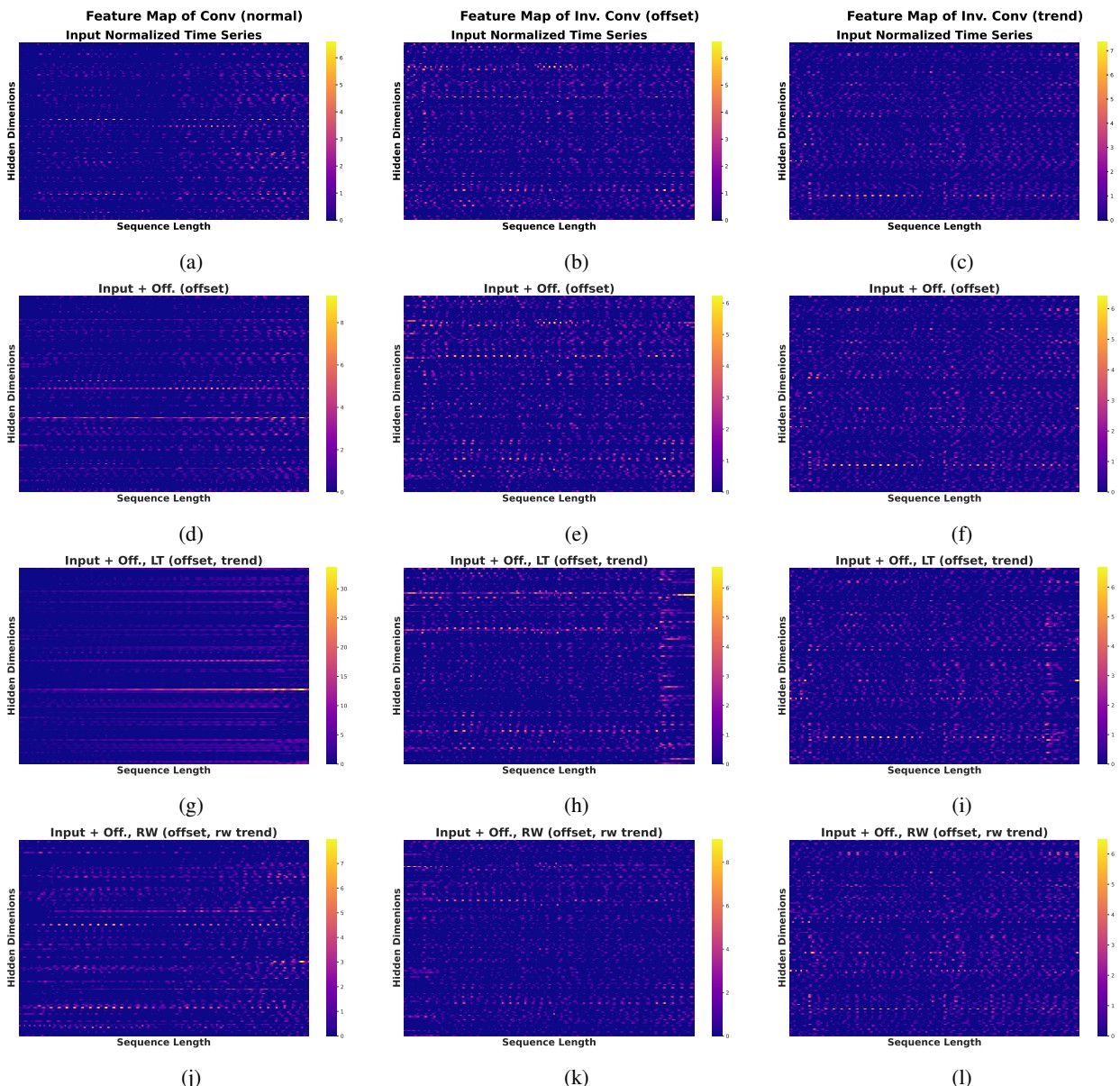

Figure 9: Comparison of the feature maps produced for the robustness study (of Table 1) on a single sample of *FordB* dataset by the different types of convolutional filters before average-pooling for the normalized input and the 3 scenarios of synthetic deformations (addition of random offset, addition of random offset and linear trend and addition of random offset and random walk trend). Each column represents convolutional filters (128 in total) of the same type; (a),(d),(g),(j) correspond to normal convolutional filters (non-invariant), (b),(e),(h),(k) correspond to offset shift invariant convolutional filters and (b),(e),(h),(k) correspond to offset shift invariant convolutional filters and (c),(f),(i),(l).

presented in Figure 8), extracted for the different considered kernel types (CONV (normal), INV. CONV (offset) and INV. CONV (trend) as heatmaps. Color in the heatmap plots corresponds to the magnitude of the activation at a specific location of the series for each hidden dimension, with a lighter color (i. e., yellow) representing higher activations.

**Deformation-Specific Activation Maps for Different Types of Filters.** The feature maps of each considered convolutional filter type with respect to invariance, i.e., normal, offset shift-invariant, and trend-invariant, reveal distinct activation patterns under different deformation scenarios.

Normal filters are highly sensitive to both offset shifts and trends, often leading to widespread activation across the entire time series (See Figures 9d, 9g). This lack of selectivity causes the model to struggle to distinguish meaningful patterns from uninformative regions, which are better captured for the plain (normalized) data ( Figure 9a). For instance, when a smooth random walk trend is introduced (Figure 9j), their activations become more uniform, indicating reduced sensitivity to meaningful variations in the input.

In contrast, offset shift-invariant filters remain robust to local shifts in the signal, exhibiting stable activation patterns even when an offset is introduced (See Figures 9b and 9e). However, their robustness is affected slightly when a linear trend is introduced (Figure 9h), as they begin to exhibit variability in their feature maps.

Finally, trend-invariant filters consistently preserve their activation patterns across all tested deformations, demonstrating resilience to both offset shifts and trends (See all Figures 9f,9i,9l). This stability ensures that relevant features are captured effectively, regardless of input distortions. Similarly, trend-invariant filters are less affected by gradual changes, maintaining consistent responses across both plain and deformed data. While normal convolutions react strongly to local variations, offset and trend-invariant filters provide stability under deformations, offering a robust representation of the underlying time series patterns in the presence of deformations.

### A.5.3. CLASSIFICATION

We next present in Figure 10, the critical difference diagram based on accuracy for the classification of *UEA* datasets. Mean rank performance is also a fair indicator of the robustness of models across several datasets and can indeed reflect consistent relative performance for the proposed method against most baselines.

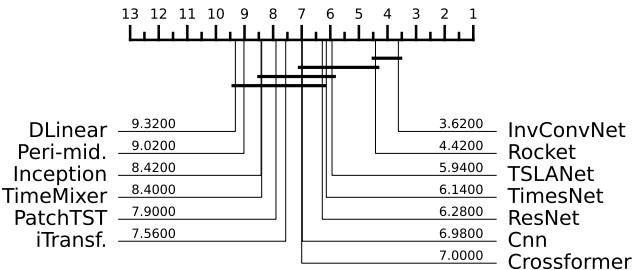

Figure 10: Critical difference diagram based on mean ranks from accuracy scores on the 26 *UEA* datasets.

INVCONVNET achieves the best mean rank across 26 UEA datasets and 13 methods. Following the implementation from the Aeon Library (Middlehurst et al., 2024), we compute mean ranks based on accuracy and assess significance using the Friedman test, which confirms the presence of statistical differences among models. Pairwise comparisons are performed based on the Wilcoxon signed-rank test with Holm correction at $\alpha = 0.1$, to identify groups of classifiers whose performance differences are not statistically significant. This procedure yields four cliques: the first includes INVCONVNET and ROCKET, while the second includes ROCKET, TSLANET, TIMESNET, RESNET, CNN and CROSSFORMER.

We also provide in Table 11 the full classification results for the 26 considered *UEA* datasets that correspond to the average of 3 runs for each combination of dataset and model. In the same table, we include again the already presented in the main paper, average accuracy for the whole collection of datasets as well as the number where each model scores first in the last row. The *Japanese Vowels* dataset is mentioned as out-of-time ('OOT') for not producing performance results since the experiment did not run within the time limits (12 hours maximum for each dataset). From the full classification results, we observe that the proposed INVCONVNET is, in several cases, slightly outperformed by the classical ROCKET method, but on

Table 11: Full Classification results for UEA datasets. Accuracy (%) is mentioned for all combinations of models and datasets. Higher is better, best methods in **bold**, second best underlined.

| Dataset | INVCONVNET | TIMESNET | PATCHTST | CROSSFORMER | ITRANSF. | PERI-MID. | TSLANET | DLINEAR | TIMEMIXER | INCEPTION | RESNET | CNN | ROCKET |
|---|---|---|---|---|---|---|---|---|---|---|---|---|---|
| *ArticularyWordRecognition* | 99.00 | 97.78 | 97.67 | 98.22 | 98.00 | 97.00 | 98.22 | 96.67 | 96.00 | 84.56 | 98.44 | 97.89 | **99.44** |
| *AtrialFibrillation* | 37.78 | 28.89 | **42.22** | 28.89 | 24.44 | 33.33 | 24.44 | 35.56 | 37.78 | 28.89 | 24.44 | 33.33 | 6.67 |
| *BasicMotions* | **100.00** | 95.00 | 70.83 | 91.67 | 86.67 | 60.83 | **100.00** | 81.67 | 72.50 | 87.50 | **100.00** | **100.00** | **100.00** |
| *Cricket* | 98.61 | 93.06 | 94.44 | 92.59 | 87.96 | 89.81 | 97.69 | 91.20 | 80.56 | 87.96 | 98.15 | 98.61 | **100.00** |
| *Epilepsy* | 95.89 | 89.61 | 97.34 | 87.44 | 69.32 | 32.13 | 96.86 | 51.45 | 64.25 | 92.27 | 94.44 | 92.27 | **98.55** |
| *EthanolConcentration* | 25.98 | 26.24 | 23.32 | **39.67** | 23.57 | 24.21 | 22.18 | 24.97 | 24.97 | 23.57 | 21.93 | 22.81 | 29.40 |
| *FaceDetection* | 64.71 | **67.50** | 64.77 | 65.26 | 65.50 | 63.14 | 56.59 | 62.97 | 64.17 | 63.88 | 54.82 | 52.75 | 59.13 |
| *FingerMovements* | **56.33** | 55.00 | 53.33 | 52.33 | 53.67 | 50.33 | 55.00 | 48.67 | 49.67 | **56.33** | 53.00 | 53.00 | 54.00 |
| *HandMovementDirection* | 40.99 | **64.41** | 47.75 | 57.21 | 45.95 | 36.94 | 45.50 | 59.01 | 51.80 | 31.08 | 36.04 | 29.28 | 44.59 |
| *Handwriting* | 53.14 | 28.67 | 26.98 | 26.39 | 22.78 | 12.71 | 48.71 | 18.71 | 25.45 | 17.22 | 37.10 | 36.20 | **56.27** |
| *Heartbeat* | **77.40** | 68.29 | 66.18 | 68.13 | 65.85 | 73.17 | 75.77 | 69.92 | 66.83 | 70.41 | 69.76 | 62.76 | 73.17 |
| *InsectWingbeat* | 'OOT' | 'OOT' | 'OOT' | 'OOT' | 'OOT' | 'OOT' | 'OOT' | 'OOT' | 'OOT' | 'OOT' | 'OOT' | 'OOT' | 'OOT' |
| *JapaneseVowels* | 97.66 | 91.71 | 94.68 | 96.76 | 96.67 | 88.83 | 96.85 | 93.33 | 94.41 | 91.80 | **98.83** | 98.38 | 97.39 |
| *Libras* | 88.70 | 79.07 | 76.11 | 86.30 | 84.63 | 86.30 | 84.81 | 50.19 | 72.41 | 57.04 | 88.70 | 91.11 | |
| *LSST* | 55.04 | 12.77 | 48.35 | 11.21 | 8.95 | 31.08 | 10.41 | 31.85 | 48.61 | 35.71 | 8.99 | 9.37 | **60.76** |
| *MotorImagery* | 49.67 | 52.00 | 50.67 | 55.00 | 51.67 | **55.67** | 47.67 | 50.33 | 49.00 | 51.67 | 51.33 | 51.33 | 46.33 |
| *NATOPS* | 95.74 | 93.33 | 75.00 | 87.41 | 83.15 | 88.70 | 94.63 | 92.78 | 76.48 | 90.74 | **96.67** | 95.74 | 87.96 |
| *PEMS-SF* | 80.35 | 78.61 | 81.89 | 84.39 | **87.86** | 63.39 | 79.96 | 80.15 | 84.01 | 75.53 | 79.38 | 74.37 | 80.15 |
| *PenDigits* | 98.78 | 98.48 | 97.52 | 97.12 | 98.35 | 96.57 | 98.12 | 87.32 | 97.48 | 97.75 | 98.70 | **98.81** | 98.08 |
| *PhonemeSpectra* | **29.82** | 14.31 | 12.62 | 12.63 | 10.36 | 13.64 | 26.71 | 6.72 | 10.05 | 21.91 | 28.66 | 27.15 | 27.69 |
| *RacketSports* | 87.72 | 82.68 | 76.75 | 79.82 | 74.12 | 79.39 | 88.16 | 67.98 | 77.19 | 83.77 | 90.57 | 84.43 | 90.35 |
| *SelfRegulationSCP1* | 86.12 | **87.60** | 78.84 | 85.32 | 87.49 | 81.46 | 79.18 | 83.39 | 83.05 | 81.57 | 80.32 | 84.64 | 84.53 |
| *SelfRegulationSCP2* | 54.44 | 48.15 | 45.19 | 47.59 | 48.15 | 51.67 | 53.89 | 45.74 | 49.26 | 53.33 | 48.15 | 48.33 | **54.82** |
| *SpokenArabicDigits* | 99.47 | 98.83 | 97.92 | 98.67 | 98.86 | 97.24 | **99.58** | 95.85 | 97.71 | 98.50 | 99.23 | 98.98 | 99.56 |
| *StandWalkJump* | 28.89 | 33.33 | 51.11 | 24.44 | 51.11 | 33.33 | 48.89 | 33.33 | **57.78** | 26.67 | 40.00 | 31.11 | 48.89 |
| *UWaveGestureLibrary* | 92.92 | 86.35 | 83.02 | 84.69 | 85.42 | 80.94 | 87.71 | 77.92 | 83.65 | 61.77 | 81.35 | 71.56 | **93.33** |
| **Avg. Accuracy (%)** | **71.81** | 66.87 | 66.18 | 66.37 | 64.42 | 60.87 | 68.70 | 61.51 | 64.60 | 62.86 | 67.37 | 65.67 | 71.29 |
| **1st Count** | 4 | 3 | 1 | 1 | 1 | 1 | 2 | 0 | 1 | 1 | 5 | 2 | 8 |

average, is among the first best-competing models for most datasets, which explains its performance superiority in terms of average accuracy for the whole *UEA*.

In several studies (Wu et al., 2022; Zhou et al., 2023), only a subset of 10 *UEA* datasets is considered, and we also present once again the results for this subset along with total the average accuracy in 12. Similar observations can be made as those for Table 11, with the proposed INVCONVNET scoring the best average accuracy of 73.22%, followed by ROCKET.

Table 12: Full Classification results for a subset of 10 UEA datasets. Accuracy (%) is mentioned for all combinations of models and datasets. Higher is better, best methods in **bold**, second best underlined.

| Dataset | INVCONVNET | TIMESNET | PATCHTST | CROSSFORMER | ITRANSF. | PERI-MID. | TSLANET | DLINEAR | TIMEMIXER | INCEPTION | RESNET | CNN | ROCKET |
|---|---|---|---|---|---|---|---|---|---|---|---|---|---|
| *EthanolConcentration* | 25.98 | 26.24 | 23.32 | **39.67** | 23.57 | 24.21 | 22.18 | 24.97 | 24.97 | 23.57 | 21.93 | 22.81 | 29.40 |
| *FaceDetection* | 64.71 | **67.50** | 64.77 | 65.26 | 65.50 | 63.14 | 56.59 | 62.97 | 64.17 | 63.88 | 54.82 | 52.75 | 59.13 |
| *Handwriting* | 53.14 | 28.67 | 26.98 | 26.39 | 22.78 | 12.71 | 48.71 | 18.71 | 25.45 | 17.22 | 37.10 | 36.20 | **56.27** |
| *Heartbeat* | **77.40** | 68.29 | 66.18 | 68.13 | 65.85 | 73.17 | 75.77 | 69.92 | 66.83 | 70.41 | 69.76 | 62.76 | 73.17 |
| *JapaneseVowels* | 97.66 | 91.71 | 94.68 | 96.76 | 96.67 | 88.83 | 96.85 | 93.33 | 94.41 | 91.80 | **98.83** | 98.38 | 97.39 |
| *PEMS-SF* | 80.35 | 78.61 | 81.89 | 84.39 | **87.86** | 63.39 | 79.96 | 80.15 | 84.01 | 75.53 | 79.38 | 74.37 | 80.15 |
| *SelfRegulationSCP1* | 86.12 | **87.60** | 78.84 | 85.32 | 87.49 | 81.46 | 79.18 | 83.39 | 83.05 | 81.57 | 80.32 | 84.64 | 84.53 |
| *SelfRegulationSCP2* | 54.44 | 48.15 | 45.19 | 47.59 | 48.15 | 51.67 | 53.89 | 45.74 | 49.26 | 53.33 | 48.15 | 48.33 | **54.82** |
| *SpokenArabicDigits* | 99.47 | 98.83 | 97.92 | 98.67 | 98.86 | 97.24 | **99.58** | 95.85 | 97.71 | 98.50 | 99.23 | 98.98 | 99.56 |
| *UWaveGestureLibrary* | 92.92 | 86.35 | 83.02 | 84.69 | 85.42 | 80.94 | 87.71 | 77.92 | 83.65 | 61.77 | 81.35 | 71.56 | **93.33** |
| **Avg. Accuracy (%)** | **73.22** | 68.20 | 66.28 | 69.69 | 68.22 | 63.68 | 70.04 | 65.30 | 67.35 | 63.76 | 67.09 | 65.08 | 72.78 |

