# OpenReview forum: "Time Series Representations with Hard-Coded Invariances"
_ICML.cc/2025/Conference — ICML 2025 poster_

### Official Review · Reviewer_MxbV · 2025-03-14

**Overall Recommendation:** 2

**Summary:**

This paper posits that invariances to the deformations are critical for time series tasks such as classification. The paper mathematically formulates the invariance in the language of group theory and further technically designs efficient and hard-coded invariant convolutions for specific deformations commonly observed in time series (scaling, offset shift, and trend). Experiments are conducted on time series classification and anomaly detection tasks to verify its effectiveness.

**Claims And Evidence:**

The claims are supported by both theoretical and empirical evidence.

**Essential References Not Discussed:**

To the best of my knowledge, all key references have been thoroughly addressed in the paper.

**Experimental Designs Or Analyses:**

The experiments are thorough, including the results of multiple tasks (classification, anomaly detection) and efficiency analysis.
However, some of the results raised my concerns:
1. In Table 1, InvConvNet is not optimal on normalized data (without additional deformation), which questions its practicality.
2. In Table 2 and Table 11, the performance improvement is not markedly significant. For instance, InvConvNet compared to the second-best baselines:
UEA 71.81 over 71.29, +0.52 (0.7%),
UCIHAR 96.63 over 96.04, +0.59 (0.6%),
Epilepsy 98.43 over 98.38, +0.05 (0.05%).

**Methods And Evaluation Criteria:**

The mathematical formulation of the problem and the design of the method are clear, and the selection of evaluation criteria is well-justified.

**Other Comments Or Suggestions:**

I suggest relocating Figure 2 to an earlier section of the manuscript, facilitating an intuitive comprehension of deformation for readers as they engage with the introduction section.

**Other Strengths And Weaknesses:**

Strengths:
1. The paper introduces a novel approach to capturing invariances from deformations in time series, which is significant for tasks such as time series classification. Technically, a novel convolution, Inv. Conv, is proposed to keep invariant to rigid deformations. It is computationally efficient through FFT.
2. As mentioned in Theoretical Claim, the mathematical formulation is rigorous and sound, building on group theory and projection operators.
3. The experiments are thorough, including the results of multiple tasks (classification, anomaly detection) and efficiency analysis. Additional results in supplementary material further improve the quality of the paper.

Weaknesses:
1. The presentation of the paper could be improved; some sections are dense, making it challenging to follow for those less familiar with group theory.
2. As mentioned in Experimental Design, in Table 1, InvConvNet is not optimal on normalized data (without additional deformation), which questions its practicality.
3. As mentioned in Experimental Design, the performance improvement is not markedly significant in Table 2 and Table 11.

**Questions For Authors:**

1. Based on Weakness 1, InvConvNet does not exhibit optimal performance on normalized data, and it only demonstrates a performance advantage when artificial deformations are introduced. However, raw data inherently possesses distribution shifts. Thus, what is the practical significance of artificially adding deformations? Furthermore, how can you ensure that the added deformations are justifiable for the specific dataset? For instance, introducing certain deformations to ECG signals might cause anomalies to the entire dataset that contravene established medical principles.
2. Time series forecasting methods are discussed in Related Work. Can invariant convolution enhance the performance of time series forecasting?

**Relation To Broader Scientific Literature:**

This work connects to broader literature by formalizing time series invariances through group actions, extending principles from image invariance to the time series domain. It draws inspiration from the traditional time series mining methods' focus on invariance (e.g., warping invariance through DTW, amplitude and offset invariances through Z-normalization) and proposes invariant convolutions for deep learning, offering a new perspective for handling deformations in time series analysis.

**Theoretical Claims:**

The mathematical formulation is rigorous and sound, building on group theory and projection operators. Specifically, propositions 1–2 are correctly proven using orthogonal projection operators, ensuring invariance to specified deformations.

---

> ### Author Rebuttal · Authors · 2025-04-01
>
> We sincerely appreciate the reviewer's time and thoughtful feedback. In the following, we thoroughly address each identified weakness and question.
>
> **[Experimental Design]**
>
> - **Performance on UCR normalized data:**
> We address your concern in Q1 below. To clarify, synthetic deformations are used only in test sets for the robustness study of convolutions (Section 4.1, Table 1). In contrast, in all other experiments (Sections 4.2–4.4 for classification and Appendix A.5.1 for anomaly detection), performance is evaluated on default data without added deformations.
>
> - **Significance of results:** We invite the reviewer to see the critical diagram difference for UEA in the rebuttal for **reviewer gvG9**.
>
> **[Other]**
>
> > The presentation of the paper could be improved [...].
>
> To enhance readability, we will move Figure 2 to an earlier section of the manuscript to better standard deformations and add a brief intuitive introduction in the method section. See our response to **reviewer Bfty** in section **[Clarity]** for details.
>
> Replies to Questions:
>
> > Q1. *InvConvNet does not exhibit optimal performance on normalized data [...]. What is the practical significance of artificially adding deformations?*
>
> In the robustness experiment ("I. Robustness to Deformations", section 4.1), the performance of invariant convolutions compared to normal ones on the plain (i.e., z-normalized input without deformations) is better for 2 out of 5 plain UCR datasets. It is still close for the rest of the cases (average -4.7\% drop for the 3 data cases). Contrary when synthetic deformations are added, the performance drops for normal convolutions are around -50\% (on average for the 4 considered deformations), whereas for the trend invariant convolution, this drop is just around -2\% (on average for the 4 considered deformations) showing the competitiveness and robustness of the proposed layer in comparison to its standard counterparts.
>
> The motivation for the robustness study on UCR datasets is as follows: Most UCR datasets, including those used in this study, are already z-normalized. Their univariate nature, limited class diversity, and relative trend stability make classification easier for normal convolutions. To systematically assess the impact of synthetic deformations, we use these datasets in a controlled setting, progressively increasing deformation complexity (from offset shifts to linear trends and smooth random walks) to evaluate how different convolutional components, including those invariant to such distortions, respond.
>
> > Q2. *How can you ensure that the added deformations are justifiable for the specific dataset?*
>
> Offset and trend invariances are common in real-world applications like PPG monitoring and ECG analysis (see reply **[Other]** to **reviewer gvG9**). They are tackled by baseline wander removal and offset correction techniques that preserve physiological signals while eliminating low-frequency noise. The closest easy-to-generate deformation, a smooth random walk, is included in the robustness experiment (see Table 1).
>
> > Q3. *Can invariant convolution enhance the performance of time series forecasting?*
>
> Our invariant layers, combined with the example decoder used for anomaly detection, naturally extend to forecasting by employing the same lightweight decoder based on linear layers applied to the learned coefficients. We next provide some preliminary results (in MSE) for the ETT-small datasets (for horizon length h=96), where we use a fixed seed and compare our approach against 8 baselines (most derived from the package Time Series Library).
>
> | Datasets (h=96) | **InvConvNet** | **TimeMixer** | **PerimidFormer** | **iTransformer** | **PatchTST** | **DLinear** | **TimeNet** | **FedFormer** | **Autoformer** |
> |-----------------|----------------|---------------|-------------------|------------------|--------------|-------------|-------------|----------------|----------------|
> | **ETTm1**       | 0.342          | **0.319**     | 0.325             | 0.345            | 0.325        | 0.347       | 0.337       | 0.365          | 0.486          |
> | **ETTm2**       | 0.193          | **0.178**     | 0.180             | 0.184            | **0.178**    | 0.195       | 0.187       | 0.194          | 0.215          |
> | **ETTh1**       | 0.426          | 0.388         | **0.377**         | 0.402            | 0.380        | 0.407       | 0.394       | 0.378          | 0.463          |
> | **ETTh2**       | 0.340          | **0.289**     | 0.322             | 0.299            | 0.312        | 0.357       | 0.330       | 0.349          | 0.343          |
>
> These results, achieved without hyperparameter tuning (in less than a week), are close to SOTA performance, highlighting their potential. Future improvements include testing hybrid architectures for the decoder (e.g., transformers) to enhance finer granularity, capture longer dependencies, and unsupervised pretraining for better generalization.

---

> > ### Comment · Reviewer_MxbV · 2025-04-07
> >
> > Thank you for your response. While some issues were addressed, my core concern remains unresolved: the limited performance improvement of InvConvNet challenges its practical significance. During my review, I carefully examined the results in the main text and appendices. In the anomaly detection task (Table 9), the proposed model only achieves the best performance on SWaT, and the improvement is marginal (92.82 vs. 92.71, a 0.001% increase). For the classification task (Table 11), the 1st count is fewer than ResNet and Rocket, and the accuracy gains in those cases are similarly modest. Although InvConv demonstrates robustness to certain deformation in Table 1, it presents suboptimal performance on normalized data. In my view, robustness evaluations based on synthetic deformation are meaningful only if the model performs well on real-world datasets. Therefore, I maintain my original rating.

---

> > > ### Author Response · Authors · 2025-04-09
> > >
> > > We sincerely thank the reviewer for their feedback on our work. We next address the concerns raised regarding the performance improvements of the example architecture InvConvNet.
> > >
> > > First, we would like to draw the reviewer’s attention to the fact that, rather than proposing a general-purpose architecture, our main goal was to **introduce versatile layers** that integrate easily into simple models and extract deformation-invariant representations **for improved robustness**, which is supported both theoretically and empirically. To illustrate this, we used simple lightweight architectures that consistently match or surpass more complex baselines across experiments.
> > >
> > > The selected UCR datasets are z-normalized, trend-stable (as shown in Fig. 7 for FordB), and large in size, making raw data classification relatively easy, resulting in similar performances between standard and invariant convolutions. However, **under artificial deformations, standard convolutions performances drop sharply** (−48\% to −59\%), **while invariant convolutions remain robust** (0\% to −6\%). **Transfer learning results in Section 4.2 further support our robustness claims**, showing **around 4\% gains** over contrastive methods and standard convolutions.
> > >
> > > For the classification and anomaly detection experiments on raw data, we next present mean rank comparisons for a fairer assessment across datasets, as they emphasize relative performance over absolute scores. This approach offers a balanced robustness evaluation and allows statistical significance analysis.
> > >
> > > ### Table: Mean Rank in Classification Accuracy (\%) for all considered datasets (UEA + 3 additional datasets) in section 4.2
> > >
> > > | Datasets                                 | InvConvNet | TimesNet | PatchTST | Crossformer | TSLANet | DLinear | Inception | ResNet | Cnn   | Rocket  |
> > > |------------------------------------------|------------|----------|----------|--------------|---------|---------|-----------|--------|--------|----------|
> > > | *UEA + UCIHAR, Sleep-EDF, Epilepsy*      | **3.00**   | 5.6964   | 6.5714   | 6.0357       | 4.8036  | 7.7143  | 6.7500    | 4.8571 | 5.7857 | **3.7857** |
> > > | **CD Value**                              | 1.80       |          |          |              |         |         |           |        |        |          |
> > >
> > > We evaluated 10 models on multivariate classification benchmarks (UEA, UCIHAR, Sleep-EDF, Epilepsy) using mean ranks based on Accuracy and assessed significance with the Friedman test (CD = 1.80, α = 0.1, Test Statistic = 55.78, p = 0.0000). The null hypothesis is rejected, confirming statistical performance differences. Our **InvConvNet achieves the best mean rank (3.00), followed by Rocket (3.79)**, with both forming the top-performing group in the Nemenyi post-hoc test.
> > >
> > > ### Table: Mean Rank in F1-score (\%) on $5$ Anomaly Detection (AD) Datasets
> > >
> > > | Datasets              | InvConvNet | TimesNet | PatchTST | TSLANet | ETSformer | FEDformer | LightTS | DLinear | Autoformer | Pyraformer | Informer | Reformer |
> > > |-----------------------|------------|----------|----------|---------|-----------|-----------|---------|---------|-------------|-------------|----------|----------|
> > > | AD Benchmark (#5)      | *4.6*      | **3.6**  | 6.0      |  5.0   | 8.0       | 8.5       | 6.4     | 5.6     | 7.3         | 7.8         | 7.6      | 7.6      |
> > > | CD Value              | 7.56      |          |          |         |           |           |         |         |             |             |          |               |
> > >
> > > We evaluated 12 models on 5 anomaly detection datasets using mean ranks based on F1-score and the Friedman test (statistic = 10.14, p = 0.5180, CD = 7.56 at $\alpha = 0.1$). Based on the p-value, no statistically significant differences were found, and all models belong to the same group. **InvConvNet ranks second (4.6), closely behind TimesNet (3.6)**, indicating competitive and robust performance against several baselines.
> > >
> > > In terms of runtime comparisons, **InvConvNet is 6.6× faster than TimesNet** and **5.3× faster than TSLANet in time per epoch** on average for the $5$ datasets. For instance, On SMD, it's 22.5× faster than TimesNet and 11.4× faster than TSLANet, with only minor performance drops (-0.56\% and -0.28\%). These time cost results align with Figure 3, where InvConvNet is 1.2× faster than TSLANet on Heartbeat (from UEA) while improving classification accuracy by 1.6\% (77.40\% vs. 75.77\%).
> > >
> > > We hope the above clarifications highlight the robustness, consistent performance, and computational efficiency of the example InvConvNet architectures, which primarily serve to evaluate our theoretically grounded invariant convolutions. We greatly appreciate your continued interest in our work and hope that our justifications further highlight the potential and impact of our proposed convolutional layers for different time series applications.

---

### Official Review · Reviewer_7HtV · 2025-03-14

**Overall Recommendation:** 3

**Summary:**

This paper proposes a novel mathematical method to consider the deformation-invariance during representation learning, which is beneficial for downstream tasks such like classification. It designs a G-variant convolution model called TS-TCC to obtain deformation-invariant embeddings, which provides robustness by decoupling the key information from deformations. Theoretically, this paper proposes using group actions to represent deformations and proves that orbit-injective embedding could map the orbit (the deformations on some time series) to the same embedding, thus avoiding the adverse effects from deformations during training. Empircally, this paper constructs the Invariant & Orbit-injective embedding through a convolution and validates its performance on the classification task.

## update after rebuttal
Thank you for the respnse, which clarify things. I will keep my relatively positive perspetive.

**Claims And Evidence:**

The paper manages to solve the deformation phenomenon in time series representation learning, which hinders the capture of key information.  To achieve this, it starts from a mathematical methodology, utilizes the group theory and mesurement theory to construct a specific convolution. The theoretical aspects are correct and can support the author's motivations.

However, the basic assumptions about the deformation phenomenon seem not strong enough. The showcase in Figure 1 conveys the information that the key information (seems to be the period) could be coupled in the trend ( a kind of deformation). While in forecasting tasks, such decomposition methods (moving average, convolutions) are widely used [1,2,3] to extract the trend. CycleNet [4] has also demonstrated that global shared periods can be easily extracted through a learnable matrix. In other words, the authors seem only consider some naive linear deformations (the equation 2 and Figrue 2), are ANNs really hard to handle these?

[1]  Autoformer: Decomposition transformers with auto-correlation for long-term series forecasting.
[2] Are transformers effective for time series forecasting?
[3] Duet: Dual clustering enhanced multivariate time series forecasting.
[4] Cyclenet: enhancing time series forecasting through modeling periodic patterns.

**Essential References Not Discussed:**

More recent studies should be discussed in the related works and added as baselines. See [5] - [6].

**Experimental Designs Or Analyses:**

It is recommended to evaluate the effects on several other downstream tasks such as forecasting and anomaly detection. The baselines are also somewhat outdated that only one work is published in 2024. The authors can take UP2ME [5],  Peri-midFormer [6] into consideration.

[5] UP2ME: Univariate Pre-training to Multivariate Fine-tuning as a General-purpose Framework for Multivariate Time Series Analysis

[6]  Peri-midFormer: Periodic Pyramid Transformer for Time Series Analysis

**Methods And Evaluation Criteria:**

Though the methodology is correct and novel, the benchmarking is somewhat weak. The paper proposes a representation method while only validating its performance on classification tasks.  Since the deformation phenomenon is ubiquitous, why not explore its performance on various downstream tasks such as forecasting, anomaly detection (should be discussed in the main text)?

**Other Comments Or Suggestions:**

none

**Other Strengths And Weaknesses:**

none

**Questions For Authors:**

see previous sections

**Relation To Broader Scientific Literature:**

Discussing some transformation problems from the perspective of group theory as in this paper is a research prospect, because mathematical methods often bring strong constraints. As long as the assumptions are correct, the probability of effectiveness will be greater.

**Theoretical Claims:**

The proofs are correct.

---

> ### Author Rebuttal · Authors · 2025-04-01
>
> We sincerely thank the reviewer for their time and thoughtful evaluation of our work. In the following section, we address any concerns raised.
>
> **[Claims and Evidence]**
>
> > "However, the basic assumptions about the deformation phenomenon seem not strong enough[...] While in forecasting tasks, such decomposition methods (moving average, convolutions) are widely used [...]"
>
> For time series forecasting, the suggested papers [1,2,3,4] assume a seasonal-trend decomposition of time series to derive suited ANN architectures. Most methods employ a (learnable) moving average kernel to infer the trend and propose different ways to deal with residual time series. The average moving kernel corresponds to a specific deformation in the proposed framework: the offset shift (see Figure 2). Assuming such offset-invariant kernels, the proposed layer offers different views of the residual time series. When focusing on trend deformations, the trend approximation could be better inferred by assuming a higher-order Taylor expansion (linear, quadratic, etc.). In the anomaly detection (Appendix A.5.1) and forecasting experiments (see rebuttal reply **Q3** to **reviewer MxbV**), we also leverage trend-residual decomposition in lightweight architectures by feed-forwarding the projections coefficients to the reconstruction layer.
> In classification, PerimidFormer (Classification Table in rebuttal response to **reviewer Bfty**) and Dlinear (Table 2 of the manuscript) are also based on a trend-residual decomposition, yet are significantly outcompetes by the proposed invariant convolutions. For instance, on the UEA datasets, our example architecture InvConvNet, performs better than PerimidFormer and Dlinear (acc: 71.81 vs (59.71 and 61.51 resp.)), suggesting that deformation-based trend invariance is better suited than standard trend-residual decomposition for classification with neural networks. Overall, the proposed layer seems more versatile and robust compared to existing architectures on several tasks while guaranteeing state-of-the-art performances.
>
> > "The authors seem only consider some naive linear deformations [...] are ANNs really hard to handle these?"
>
> Standard ANNs often struggle with generalization on noisy, non-stationary time series, as illustrated in Figure 1, where a standard CNN fails to capture relevant ECG features. To test whether CNNs can learn invariance, we conduct a robustness experiment (Table 1), showing that classification performance declines as datasets are progressively deformed. Standard convolutions prove sensitive to spatiotemporal distortions (avg. acc. drop: -50\%), whereas hard-coded invariant convolutions offer better generalization  (avg. acc. drop: -2\%) than contrastive learning (avg. acc. drop: -23\%). Figure 8 (Appendix) further visualizes this, revealing that standard CNN feature maps become distorted, while invariant convolutions retain structured representations (see Figure 7 for corresponding FordB dataset deformations). These findings align with (Kvinge et al., 2022) for image data. Finally, in transfer learning (Table 4), invariant convolutions surpass standard architectures and contrastive frameworks by at least 4\%.
>
> - Kvinge, H., Emerson, T., Jorgenson, G., Vasquez, S., Doster, T., \& Lew, J. (2022). In what ways are deep neural networks invariant and how should we measure this?. NeurIPS, 35, 32816-32829.
>
> >The paper proposes a representation method [...] tasks such as forecasting, anomaly detection?
>
> We conduct extensive experiments across diverse real-world applications to validate our theoretical framework of time series invariant convolutions, leveraging common offset and shift invariances in shallow (Figures 5 and 6) and lightweight example architectures (Figure 3). We have demonstrated robustness against offset and trend deformations (Section 4.1), strong performance in multivariate classification (Section 4.2), and transfer learning for classification (Section 4.3). Additionally, we extend our analysis to anomaly detection in Appendix A.5.1. Rather than proposing a general-purpose architecture, targeted to specific time series tasks, our goal was to develop versatile layers that integrate seamlessly with existing methods to enhance robustness. To illustrate this, we employ lightweight architectures—such as linear decoders for regression—that, despite their simplicity, match or surpass complex baselines. While our current focus is classification and anomaly detection, our approach naturally extends and holds promise for forecasting, as noted in our response and experiments provided to **reviewer MxbV** (rebuttal reply **Q3**).
>
> **[Experimental Designs Or Analyses]**
> We show improved performance against the suggested baselines UP2ME [5] and PerimidFormer [6]. Please refer to the Tables in the rebuttal reply to **reviewer Bfty**. We will also include a discussion about the additional baselines in the revised version of the manuscript (not added here due to the word limit).

---

### Official Review · Reviewer_Bfty · 2025-03-14

**Overall Recommendation:** 3

**Summary:**

The paper proposes convolutional neural network operations explicitly designed with hard-coded invariances (e.g., scaling, offset shift, linear trends) for improved time series representation learning. By formulating invariances through group theory and embedding them directly into convolutional layers, the authors show empirically that their method enhances robustness against common temporal deformations, achieves competitive or superior accuracy across benchmark classification tasks, and offers computational efficiency advantages over learned invariances or standard CNN approaches.

## update after rebuttal
Most of my concerns have been addressed. I will keep my score to support the paper.

**Claims And Evidence:**

The claims presented in the paper are generally supported by clear and convincing evidence. The authors provide a thorough mathematical formulation of their hard-coded invariant convolutions and validate the theoretical properties empirically on multiple tasks, including synthetic deformation robustness tests, benchmark classification datasets, and transfer learning scenarios. The experimental setup is extensive and compares against relevant state-of-the-art baselines, clearly supporting the claims regarding improved robustness and competitive accuracy. However, while the authors claim computational efficiency benefits through FFT-based convolutions, additional explicit runtime comparisons or complexity analyses against more baseline architectures (especially in larger-scale or real-world scenarios) would further strengthen this claim.

**Essential References Not Discussed:**

To my best knowledge, essential references are discussed.

**Experimental Designs Or Analyses:**

I have checked the soundness/validity of experimental designs and analyses. The experimental designs and analyses presented in the paper appear sound and valid. Specifically, the robustness evaluation involving synthetic deformations (offset shifts, linear trends, random walks) provides clear evidence to support the claimed invariances. Benchmarking against multiple state-of-the-art baselines across diverse datasets from the widely used UCR and UEA archives ensures rigorous comparative assessment. The transfer learning experiments are also well-designed, clearly illustrating the generalization capabilities of the proposed method. One minor area for improvement could be providing additional details on hyperparameter tuning and statistical significance testing for the observed improvements to further reinforce the validity of the conclusions.

**Methods And Evaluation Criteria:**

The proposed methods and evaluation criteria are generally appropriate. But I think authors should include some more stronger baselines ([1] [2] etc.). Also, I suggest authors to provide more visualization results to demonstrate the effectiveness of capturing invariances in time series.

[1] iTransformer: Inverted Transformers Are Effective for Time Series Forecasting
[2] TimeMixer

**Other Comments Or Suggestions:**

I am willing to raise my score if authors could address my concerns.

**Other Strengths And Weaknesses:**

Strengths:
* Originality: The paper demonstrates clear originality by creatively combining established mathematical concepts (group invariances) with convolutional neural network architectures, leading to explicit and computationally efficient invariant representations for time series.
* Significance: The contributions are significant, addressing important practical challenges in the robustness of deep learning models for real-world temporal data.

Weaknesses:
* Clarity: the mathematics framework is hard to understand. It would be better if authors could give some intuitive explanations.

**Questions For Authors:**

I do not have other questions.

**Relation To Broader Scientific Literature:**

The paper's key contributions closely relate to recent research in deep learning for time series, particularly efforts to enhance model robustness via invariant representations. Specifically, the authors build upon concepts from group theory and invariant/equivariant neural networks, aligning with prior research on hard-coded invariances in domains such as images and graphs (e.g., translation invariance in CNNs, permutation invariance in GNNs). Unlike prevalent methods that introduce invariances implicitly through data augmentation and contrastive learning (e.g., TS-TCC, TS2Vec), this paper explicitly incorporates invariances into convolutional architecture design, extending established ideas (e.g., ROCKET's random convolutions) with rigorous theoretical foundations. Thus, the presented method bridges prior theoretical findings about invariance modeling with practical CNN-based approaches widely adopted in time series classification, positioning itself clearly and convincingly within the broader scientific landscape.

**Theoretical Claims:**

I checked the correctness of the theoretical claims presented, specifically focusing on the mathematical formulations related to group invariances and the definition of invariant convolution operators (Section 3, including Propositions 1 and 2). The proofs provided in Appendix A.1 (referred to clearly within the paper) appear mathematically sound and correctly derived, relying appropriately on established concepts from group theory and functional analysis. But I am not very familiar with this field, so I am not very confident about my judgement.

---

> ### Author Rebuttal · Authors · 2025-04-01
>
> We sincerely appreciate the reviewer's time and effort in evaluating our work. Below, we provide our responses to the main suggestions.
>
> **[Claims And Evidence]**
>
> > "authors claim computational efficiency benefits [...], additional explicit runtime comparisons [...]".
>
> We have demonstrated the computational efficiency of the proposed invariant layers over baselines, with memory complexity and training time comparisons shown on the UEA Heartbeat dataset (with 61 channels and 405 timestamps) in Figure 3. Similar time comparisons on larger datasets will be included in the revised manuscript. More details on the fast computation of our convolutional layers via FFT can be found in Section 3.2 (pages 4-5).
>
> **[Methods and Evaluation Criteria]**
>
> - **Additional Experiments.**
> We next present comparisons for classification on the 26 UEA datasets and the 5 anomaly detection (AD) datasets, evaluating the additional methods: TimeMixer, iTransformer, Peri-midFormer, and UP2ME
>
> ## Additional Results Classification Acc.(\%)
>
> | Datasets            | **InvConvNet** | TimeMixer | iTransformer | PerimidFormer | UP2ME  |
> |---------------------|----------------|-----------|--------------|---------------|--------|
> | **UEA** (26 datasets) | **71.81 ± 0.80** | 64.60 ± 1.59 | 64.42 ± 2.05 | 59.71 ± 3.03 | 54.57 ± 3.88 |
>
> ## Additional Results Anomaly Detection
>
> | Datasets  | **InvConvNet** | TimeMixer | iTransformer | PerimidFormer | UP2ME (4/5 datasets) |
> |-----------|-----------------|-----------|--------------|---------------|----------------------|
> | **SMD**   | **84.05 ± 0.16** | 83.20 ± 0.06 | 82.38 ± 0.99 | 83.34 ± 0.63  | **84.34 ± 0.13** |
> | **MSL**   | **80.68 ± 0.01** | 67.12 ± 3.33 | 72.66 ± 0.04 | 80.93 ± 0.06  | 80.66 ± 0.01 |
> | **SMAP**  | **68.29 ± 0.07** | 65.55 ± 0.32 | 66.86 ± 0.08 | 67.62 ± 0.02  | 67.56 ± 0.96 |
> | **SWaT**  | **92.82 ± 0.19** | 91.77 ± 1.30 | 92.68 ± 0.01 | 92.17 ± 0.05  | OOM |
> | **PSM**   | 96.34 ± 0.01     | 94.01 ± 0.77 | 95.15 ± 0.14 | 96.31 ± 0.09  | **96.42 ± 0.01** |
> | **Avg. F1 (\%)** | **84.44 ± 0.09** | 80.33 ± 1.16 | 81.95 ± 0.25 | 84.07 ± 0.17  | 82.24 ± 0.28 |
>
> Our method consistently retains its advantage in both classification and anomaly detection tasks.  We were unable to obtain results for the SWaT dataset using UP2ME (marked with out-of-memory), even after reducing the hyperparameters to manage memory usage.
>
> **Implementation Details:** Based on their official GitHub implementations, only iTransformer has been adapted for classification and anomaly detection. Optimal hyperparameters are provided for iTransformer on 10 UEA datasets and all AD datasets, while for TimeMixer, we used default values for non-forecasting tasks. Hyperparameters for Peri-midFormer are sourced from its GitHub for classification (on 10 UEA datasets) and anomaly detection tasks. Additionally, we test the backbone of UP2ME. We adapted TimeMixer and UP2ME for classification by reshaping the encoder outputs into a one-dimensional vector and passing them through fully connected layers for predictions.
>
> - **Visualizations of Capturing Invariances.**
> We will definitely include additional visualizations of the invariances captured by our layers in the revised manuscript. For now, we have included visualizations of offset and trend deformations on UCR datasets, along with feature maps for normal and invariant kernels, in Figures 7 and 8 of the Appendix.
>
> **[Experimental Designs Or Analyses]**
>
> We will incorporate the reviewer's suggestions in the revised manuscript. We already provide extensive details on hyperparameter tuning for the proposed method and the baselines in **Section A.4 Implementation Details**. While results are currently marked based on mean performance and std overlaps, we will consider paired statistical t-tests for all datasets and best methods (not provided due to time constraints). Please also refer to the critical diagram difference for UEA in the rebuttal for **reviewer gvG9**.
>
> **[Clarity]**
>
> > "The mathematics framework is hard [...] intuitive explanations."
>
> To enhance readability, we will relocate Figure 2 to an earlier section of the manuscript to clearly showcase standard deformation types. The methods section will also add a short, intuitive introduction: "The proposed mathematical framework consists of two main components. The first is a group action that formalizes how certain deformations transform time series, resulting in their deformed counterparts. Only deformed time series are observable in practice, such as those influenced by noise or trends. However, embeddings that remain invariant to these deformations are essential for many applications. To address this, the second component is a mapping function that constructs embeddings of deformed time series while remaining invariant to specific deformations. Finally, these embedding maps can be efficiently integrated into convolutional operations to extract robust local features to non-informative deformations.

---

> > ### Comment · Reviewer_Bfty · 2025-04-04
> >
> > Thanks for your response. Most of my concerns have been addressed. I will keep my score to support the paper.

---

> > > ### Author Response · Authors · 2025-04-08
> > >
> > > Dear reviewer Bfty,
> > >
> > > We would like to thank you for your positive feedback on our rebuttal replies. We are grateful for the opportunity to address your concerns, and we are pleased that we have managed to resolve most of the issues raised.
> > >
> > > The final version of the manuscript will incorporate the changes and clarifications you suggested, as presented in our rebuttal replies. Specifically, we will include:
> > > *(i)* **Additional runtime comparisons** (beyond the existing ones) between the example architectures and baselines and **visualizations of the learned invariances** for more datasets,
> > > *(ii)* **Performance diagrams with statistical testing** on closely performing methods (such as the one presented in the rebuttal on UEA),
> > > *(iii)* **Results for the four additional recommended baselines** for both classification and anomaly detection, and
> > > *(iv)* **A simple roadmap for the presented mathematical framework** and the considered deformations by relocating relevant figures and providing a more intuitive introduction to the method for a more general audience.
> > >
> > > We believe that the above revisions indeed strengthen the manuscript, improving its clarity and impact. We hope *these improvements*, along with *the overall theoretical and methodological contribution of the work*, will help reinforce your positive assessment of the submission even further. We remain happy to provide any last-minute clarification if necessary.
> > >
> > > Once again, thank you for your valuable feedback and continued support of our research.

---

### Official Review · Reviewer_gvG9 · 2025-03-19

**Overall Recommendation:** 4

**Summary:**

The article introduces a mathematical framework for integrating invariant into convolution operators for time series. The scaling, offset shifting, and trend invariant are particularly studied. A large number of experiments are then conducted, demonstrating the advantages of these convolutions in terms of their robustness to distortions, their performance in classification, as well as their effectiveness in transfer learning and anomaly detection.

## update after rebuttal
Given the responses provided during the rebuttal, I am increasing my score.

**Claims And Evidence:**

Numerous experiments support the authors' claims regarding the advantage of incorporating hard-coded invariants into convolutions. A first series of experiments artificially introduces distortions to demonstrate the robustness of their method compared to state-of-the-art approaches (Table 1), while comparing results across different types of invariance and convolution. An ablation study is also conducted on classification results to highlight the role of each component (Table 3). Finally, transfer learning experiments are carried out to demonstrate the relevance of the approach in this context.

**Essential References Not Discussed:**

NA

**Experimental Designs Or Analyses:**

See above.

**Methods And Evaluation Criteria:**

The approach of evaluating the proposed model in two stages is sound, first by introducing artificial distortions and then testing it in real-world conditions. The chosen datasets are standard in the community but known to be simple. The comparison of training times between the different models is also a good aspect. The authors could have produced a synthetic diagram of the model's performance compared to others, such as a mean rank, as is standard in the field. It would have certainly shown slightly lower performance of the proposed model compared to Rocket, as indicated in the exhaustive results table (Table 11) at the end of the appendix, but it could have provided clearer insight for the reader

**Other Comments Or Suggestions:**

NA

**Other Strengths And Weaknesses:**

It is unfortunate to limit the study to trend and offset invariants, which can be handled by other approaches. The authors could have discussed other types of invariants to broaden the relevance of their work.

**Questions For Authors:**

NA

**Relation To Broader Scientific Literature:**

"To my knowledge, the most important papers in the field are well cited. The authors make a clear effort to position their work in the introduction relative to the state of the art and existing research on invariants for time series. Their proposed idea goes somewhat against the current trend—explicitly designing constraints to integrate into convolutional kernels, whereas the field generally focuses on developing models that can handle invariance in a more generic way. Consequently, the impact of this work may be limited, as there are few real-world cases where invariances are so explicitly defined in my opinion.

**Theoretical Claims:**

The article is based on the introduction of an action group for time series. I have read the formalism in detail in the main body of the article and skimmed through the appendices. The formalism is well described and requires some mathematical concepts, not all of which are explicitly stated but remain at a modest level. The examples are well chosen to illustrate the most formal parts.

---

> ### Author Rebuttal · Authors · 2025-04-01
>
> We would like to thank the reviewer for the time spent to evaluate our manuscript. We next reply to their key suggestions and comments on our work.
>
> **[Evaluation Criteria]**
>
> >"The authors could have produced a synthetic diagram of the model's performance [...]."
>
> We appreciate the reviewer's suggestion and will add a critical difference diagram, along with clarifying Rocket's first-count performance in the main section. As shown, InvConvNet achieves the best mean rank across 26 UEA datasets and 9 baselines. Using the aeon library [1], we compute mean ranks based on accuracy, confirming significance via a Friedman test (critical difference value of 1.6784 at $\alpha = 0.1$). The Nemenyi post-hoc test identifies three statistically distinct groups: 'InvConvNet' and 'Rocket' in the first, followed by 'TSLANet', 'ResNet', 'TimesNet', 'Cnn', 'Crossformer', and 'PatchTST' in the second.
>
> ## Mean Rank for UEA Classification Acc.
>
> | Datasets             | **InvConvNet** | TimesNet | PatchTST | Crossformer | TSLANet | DLinear | Inception | ResNet | Cnn | **Rocket** |
> |----------------------|----------------|----------|----------|-------------|---------|---------|-----------|--------|-----|-----------|
> | *UEA* (26 datasets)  | **3.16**       | 5.38     | 6.44     | 6.10        | 4.96    | 7.44    | 6.96      | 5.12   | 5.76| **3.68**   |
> | *CD Value* = 1.6784  |                |          |          |             |         |         |           |        |     |           |
>
> [1] https://www.aeon-toolkit.org/en/latest/examples/benchmarking/published_results.html#References
>
> **[Broader Scientific Relations]**
>
> >"Their proposed idea goes somewhat against the current trend [...] few real-world cases were invariances are so explicitly defined in my opinion."
>
> Recent deep learning trends for time series rely on implicit invariances from augmentations using contrastive learning, often lacking generalization guarantees (see Table 1) and incurring high computational costs. Our approach directly encodes common time series invariances within the network, ensuring both efficiency and generalization (see Figure 3, Tables 1 \& 3). Our convolutional design is rooted in group-theoretic modeling of time series deformations, which is a novel approach in this domain. Similar formalisms have introduced robust architectures for computer vision and graph neural networks. Our layers, coupled with lightweight modules (Figures 5, 6), can achieve robust and competitive performance, even on smaller datasets.
>
> **[Other]**
>
> >"It is unfortunate to limit the study to trend and offset [...] discussed other types of invariants."
>
> Baseline removal is common in applications for physiological signals, such as EEGs and PPGs, where baseline wander can mask critical variations, and removing trends can enable more accurate feature extraction [1,2]. The closest easy-to-generate deformation is a smooth random walk, which we incorporate in the robustness experiment as the most complex deformation following simple offset shift and linear trend (see feature maps visualizations in the last row of Figure 8 for smooth random walk deformation of the FordB dataset in Figure 7). Offset shifts and linear trends often arise due to sensor drift, calibration errors, or gradual measurement degradation, leading to misleading variations in the data.
> While z-normalization removes global offsets, it fails to address local distortions. Our model learns local invariance to offset shifts and trends, making it effective for real-world data. This justifies our choice of offset shift and linear trend as key deformations to experimentally validate the broader theoretical framework of time series invariant layers under group actions.
>
> In the introduction (paragraphs 4–5, pages 1–2), we discuss time series invariances, including time-shift, time-rescaling, and contrastive learning-based transformations. Commonly addressed deformations include amplitude scaling, offset shifts, and trends, tackled via z-normalization or contrastive frameworks (e.g., TS-TCC). We introduce a formal mathematical framework for time series invariance to spatiotemporal deformations, providing exact formulations of invariances rather than relying on approximations.
>
> We restrict invariant convolutions to simpler but common affine deformations, allowing invariance to trend when it can be assumed linear at the scale of the convolution kernel size. However, we plan to explore approximating the trend with non-linear functions, such as splines or higher-degree polynomials, and incorporating seasonal components with low-frequency cosine bases in future work.
>
> [1] Kaur, M., Singh, B., \& Seema. (2011). Comparing approaches for baseline wander removal in ECG signals. Proc. Int. Conf. \& Workshop on Emerging Trends in Technology, 1290-1294.
>
> [2] Awodeyi, A. E., Alty, S. R., \& Ghavami, M. (2014). On filtering photoplethysmography signals. IEEE Int. Conf. Bioinformatics \& Bioengineering, 175-178.

---

> > ### Comment · Reviewer_gvG9 · 2025-04-05
> >
> > Thanks for the clarifications, I maintain my score and support acceptance.

---

> > > ### Author Response · Authors · 2025-04-08
> > >
> > > Dear Reviewer gvG9,
> > >
> > > Thank you very much for your thoughtful feedback and the positive evaluation of our work! We are grateful for your support. Your comments have significantly helped to improve the presentation and clarity of our contribution.
> > >
> > > Based on our rebuttal, we are committed to incorporating your suggestions by:
> > > *(i)* **Adding performance diagrams** for the proposed method and baselines to further highlight the significance of performance improvements, particularly for the large number of datasets we consider,
> > > *(ii)* **Extending the discussion on practical applications** where invariance to the considered deformations is crucial, and
> > > *(iii)* **Including a discussion on additional complex deformation types** that can be captured by convolutional layers with proper construction (e.g., locally non-linear trends).
> > >
> > > We will be grateful for your consideration of these improvements and that of the **additional experiments** in the rebuttal (can be found in reply to **reviewer Bfty**), along *with the overall novelty and multiple contributions of our study* in your final evaluation score.
> > >
> > > Once again, thank you for the time you devoted to evaluating our work and for your support.

---

### Decision · Program_Chairs · 2025-05-01

**Decision:**

Accept (poster)

**Comment:**

The paper proposes to encode invariances to certain groups (e.g., scaling, shift, etc.) directly in a convolutional architecture, in order to endow the deep network with robustness to such prevalent deformations.

All in all, the paper was well received, and three out of four reviewers are positive. In general, most reviewers appreciated the soundness of the theoretical contribution, the experimental design, and the empirical results. Moreover, the rebuttal was effective and helped to increase scores.

The AC believes that this paper can have a positive impact in the field of time-series analysis and recommends Accept.

Remark: In the camera-ready version, the authors may want to mention other types of deformations, even if these are not included in their set of invariances. For example, several recent works in NeurIPS, ICML, and other venues focus on diffeomorphisms as a way of modeling nonlinear time warping in time-series data.